# FROM SEQUENCES TO SCHEMAS: HOW RECURRENT NEURAL NETS LEARN TEMPORAL ABSTRACTIONS

## ABSTRACT

A fundamental challenge in neuroscience is to understand how neural systems extract and represent abstract structure from complex, time-varying input. From language and music to action planning and sensory prediction, behavior relies on the ability to recognize relational patterns in sequences. Yet it remains unclear how such abstract temporal schemas are learned and encoded in neural population dynamics. Here, we show that training recurrent neural networks (RNNs) on an abstract sequence classification task drives the emergence of internal representations that express the underlying hierarchical structure and are supported by low-rank recurrent dynamics, whereas training on a standard next-token prediction task does not. Using sequences generated by a binary branching tree to instantiate abstract structure, we trained RNNs to classify sequences based on their abstract class (e.g., `aab`, `aad` → `AAB`; `aba`, `aca` → `ABA`), providing a label only at the end of the sequence. Despite the absence of explicit supervision at the transition level, the networks developed low-dimensional, linearly separable internal representations, reflecting the underlying hierarchical tree structure of the data, and encoding information about the sequence's path through this structure. This enables generalization across different token instantiations of the same abstract pattern. In contrast, RNNs trained on a next-token prediction task fail to form such organized dynamics or recover the underlying tree structure. However, through transfer learning, we show that when initialized with weights from a classification-trained network, prediction models learn faster and generalize better. These findings demonstrate that task objectives critically shape internal representations and that abstract structure, once learned, can serve as a reusable scaffold for diverse temporal computations.

## 1 INTRODUCTION

The ability to infer rules from specific experiences is a cornerstone of intelligent behavior, allowing organisms to make predictions and adapt to novel situations. Many real-world processes unfold as structured sequences of events, and brains have evolved to detect and exploit these sequential regularities across multiple levels of abstraction: from simple transitions and timing, to chunking, ordinal structure, and abstract, rule-based patterns, often organized into nested tree schemas [5]. Despite its importance, the precise brain mechanisms supporting abstraction and generalization remain elusive [14; 15]. A key aspect of this ability is extracting temporal regularities– particularly algebraic patterns such as `AAB` or `ABA` that capture abstract relational structures independent of sensory identity (e.g., sounds, colors, shapes, etc.) [11]. This capacity spans species: humans [23; 26], and non-human primates [23], rodents [14; 22], and birds [16; 19; 13] show sensitivity to abstract temporal patterns. How do neural circuits learn and represent such abstract temporal schemas [5]? What computational principles allow the brain to detect and generalize temporal structures across stimuli and contexts?

Recurrent Neural Networks (RNNs) are well suited to modeling time-evolving processes like working memory, context integration, and decision making [9], and are increasingly used to study how neural dynamics support behavior [1; 27; 6]. Here, we use RNNs to investigate the emergence of abstract temporal structures from sequential data, and to identify circuit-level mechanisms that enable generalization.

We systematically generate algebraic sequences using a binary tree, with terminal nodes defining abstract classes (denoted by capital letters in this manuscript, e.g., AAB or ABA) of a certain length, and individual letters (tokens) sampled from an alphabet of a given size. We train RNNs to classify sequences according to their overall abstract pattern, receiving only the class label at the end of the sequence (e.g., AAB for `aac` and `bba`; ABA for `aca` and `ada`), without any supervision on intermediate transitions.

We show that a discrete-time RNN trained on this task generalizes well to novel sequences across various lengths. Principal Component Analysis (PCA) of the hidden activity reveals low-dimensional, linearly separable representations clustered by abstract class. These clusters reflect relational structure – e.g same vs different token transitions are linearly separable–and trace out trajectories aligned with the tree structure as the sequence unfolds.

To understand the origin of this structure, we analyze the recurrent weights and show that the network dynamics are low-dimensional due to low-rank recurrent connectivity. This low-rankness constrains the neural trajectories to a low-dimensional manifold, where abstract sequence classes are linearly separated by hyperplanes aligned with leading singular vectors. Moreover, projections of the hidden activity onto these dimensions recapitulate the hierarchical structure of the generative tree, indicating the network internalizes the abstract schema through its dynamics.

We then test whether this structured representation is reusable in other tasks. Training an RNN on next-token prediction (e.g. AAB- to be completed as AABA) does not spontaneously form such low-dimensional, abstract representations. However, when initialized with weights from the classification-trained network, it learns faster and generalizes better. This suggests that the classification learned low-rank scaffold can facilitate other forms of temporal abstraction.

Together, our findings show how task constraints shape abstract sequence representations in RNNs and offer a mechanistic account – via low-rank recurrent dynamics – for how neural systems may internalize and exploit hierarchical temporal structure. These insights provide a computational framework for understanding abstraction and generalization in both artificial and biological networks.

## 1.1 PREVIOUS WORK

Understanding how neural systems represent and generalize sequential structure has been a long-standing goal in both neuroscience and machine learning. Early work focused on mechanisms for maintaining and recalling of ordered information, especially in working memory. For example, Botvinick et al. [3] proposed a neural network in which item and rank information are combined to account for serial recall behavior. More recently, Chu et al. showed that RNNs trained on ordinal prediction tasks can learn representations that generalize positional structure independently of specific inputs, suggesting that temporal abstraction can emerge through task-driven learning [4]. Wu et al. have leveraged chunking to segment high-dimensional neural population dynamics of trained networks into interpretable units that reflect underlying concepts, including structural schema [25].

Another line of work has explored augmenting RNNs with external memory systems, such as associative memory or Hebbian plasticity. These models can act as "cognitive maps" of episodic transitions that represent sequences as trajectories over latent states [24]. However, they often rely on Markovian or locally learnable transitions. Their ability to generalize to non-adjacent or higher-order dependencies–critical for abstraction–remains unclear.

From a dynamical systems perspective, several studies have investigated how low-dimensional neural activity and low-rank connectivity shape computation in RNNs [12; 18]. Mastrogiuseppe et al. have shown that low-rank recurrent connectivity confines activity to low-dimensional manifolds that support specific tasks [12]. Our work builds on this insight, showing that abstract, low-dimensional representations can emerge from low-rank recurrent connectivity and that this structure can be exploited in transfer learning scenarios.

In contrast to these previous approaches, we focus on learning abstract relational patterns such as ABAB or AABB that generalize across token identities. Rather than relying on surface-level repetition, order, or transition statistics, our task requires recognizing the generative rule behind each sequence. We show that such abstraction can emerge spontaneously in standard RNNs trained on only end-of-sequence labels, without architectural biases or item-specific encoding. Furthermore, we demonstrate that these representations are supported by low-rank recurrent dynamics and can be

reused via transfer learning to accelerate learning in more difficult prediction tasks. Together, our findings offer a mechanistic account of relational abstraction in recurrent circuits, both artificial and biological.

## 2 RESULTS

Throughout this work, we consider different neural network models to solve different tasks. Each of these models has a standard discrete-time RNN as a core component. The inputs to the RNN, denoted $\boldsymbol{x} \in \mathbb{R}^{\alpha}$, are a one-hot encoding of the elements in the sequence over letters in the alphabet. The $\mu$-th sequence, $\boldsymbol{X}^{\mu}$, with $\mu = 1 \ldots p$, is given by the elements $\boldsymbol{x}_t^{\mu}$, for $t = 1 \ldots L$. Upon processing this sequence, the hidden activity $\boldsymbol{h}^{\mu} \in \mathbb{R}^N$, evolves according to

$$
\begin{aligned}
\boldsymbol{z}_t^{\mu} &= \boldsymbol{W}_{\mathrm{h}} \cdot \boldsymbol{h}_{t-1}^{\mu} + \boldsymbol{W}_{\mathrm{in}} \cdot \boldsymbol{x}_t^{\mu} + \boldsymbol{b}_{\mathrm{h}} , \\
\boldsymbol{h}_t^{\mu} &= \phi\big(\boldsymbol{z}_t^{\mu}\big) .
\end{aligned}
\tag{1}
$$

where $\boldsymbol{W}_{\mathrm{h}} \in \mathbb{R}^{N \times N}$ is the recurrent weight matrix, $\boldsymbol{W}_{\mathrm{in}} \in \mathbb{R}^{N \times \alpha}$ is input weight matrix, $\boldsymbol{b}_{\mathrm{h}} \in \mathbb{R}^N$ are constant biases and $\phi$ is a non-linear function applied unit-wise. Here, we choose to work with rectified-linear units, i.e. $\phi(z) = \max\{0, z\}$. In the following, we will omit the sequence index unless necessary. All networks are trained via gradient-based optimization (Adam [8]) using backpropagation through time and performing mini-batch updates. Different tasks require different targets and loss functions, and the models used for different tasks only differ in the readout (see SI Sec. A. for more details).

### 2.1 RNNS CAN CLASSIFY SEQUENCES WITH REGULARITY ACROSS RANGE OF PARAMETERS

We first examine a sequence classification task in which the network is trained to classify sequences based on their underlying temporal structure (see SI Sec. B. for details on the generative process for constructing sequences). For example, sequences such as abbb and addd are generated from the class ABBB, while abba and cddc belong to the class ABBA (Fig. 1A). The network receives a label only at the end of the sequence, and the loss function is the cross-entropy between the output class probabilities generated by the network and the ground truth (see SI Sec. A.). In the remainder of the main manuscript, we focused on networks initialized in the "lazy" regime [7], where initial weights prior to training are drawn from a uniform distribution with a standard deviation of $N_{\mathrm{input}}^{-1/2}$; we have replicated all notable results in the "rich" regime in Fig. S5.

We find that sufficiently large networks achieve near zero loss on both training and test sets, indicating strong generalization performance (Fig. 1C and Fig. S1). Also, the number of classes that can be reliably generalized scales with network size, with larger networks required for generalizing a larger number of classes (Fig. S1, right).

In order to better understand the types of representations supporting generalization, we applied Principal Component Analysis (PCA) to the hidden activities at the end of the sequence. For sequences of length $L = 4$ and number of classes $C = 4$, the top three principal components account for $\sim 90\%$ of the variance, revealing a highly structured, low-dimensional geometry. Moreover, the hidden activities strongly cluster by class, and are linearly separable according to transition types: whether the sequence transitions to the same or a different letter (Fig. 1D, left). When we projected the hidden activities onto the same reduced space, as the sequences unfold, we observed a branching pattern that mirrors the hierarchical tree structure used to generate the sequences (Fig. 1D, right). These results suggest that the RNN constructs internal representations that compress and organize sequence information in a way that reflects the abstract structure of the task. Next, we investigate the geometry of these unfolding representations, and the recurrent mechanism that supports it.

### 2.1.1 LOW-DIMENSIONAL DYNAMICS REFLECTS LOW-RANK RECURRENT CONNECTIVITY

We begin by analyzing the dimensionality of the hidden activity in the classification-trained RNN as a function of time. To do so, we apply Singular Value Decomposition (SVD) to the hidden activities after each item in the sequence is presented. We define the dimensionality of this hidden activity at each time step as the number of singular vectors required to capture at least 90% of the mean squared activity. We find that the hidden activity lies in a consistently lower dimensional space compared

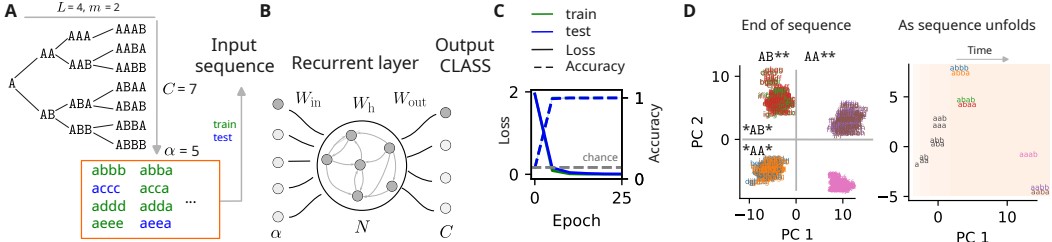

Figure 1: **Emergence of abstract sequence representations in recurrent networks. A.** Sequences are generated from a binary branching tree (example shown for sequence length $L = 4$, and $m = 2$ unique letters, corresponding to the number of branches from the root), yielding $C = 7$ abstract classes. Each class is instantiated by replacing the $m$ unique letters with symbols drawn from an alphabet (of length $\alpha = 5$). **B.** An RNN processes sequences one token at a time and is trained to classify them according to their abstract structure, receiving a class label only at the end. **C.** Training loss and accuracy across epochs (mean +/- s.d. across simulations). **D.** PCA of hidden activities at the end of training and end of sequence. Left: final hidden states cluster by abstract class and are linearly separable according to transition to same/different letter. Right: trajectories of hidden activity for one example sequence per class (indicated in different colors) plotted for each time-point as the sequence unfolds. PC2 and 3 capture branching dynamics that reflect the hierarchical structure of the generative tree as the sequence unfolds.

to the input space, and that dimensionality tends to decrease over time (Fig. 2A). This decrease in dimensionality is class-dependent: dimensionality increases with a transition to a different letter, but decreases towards the end of the sequence (Fig. 2A). This progressive dimensionality reduction suggests that the recurrent weight matrix $\boldsymbol{W}_{\mathrm{h}}$ is effectively low-rank, consistent with [12], up to a noisy term that reflects randomness in initial conditions and/or the training protocol (Fig. 2B)

$$\boldsymbol{W}_{\mathrm{h}} = \sum_{\rho=1}^{r} S_\rho \, \boldsymbol{l}_\rho \, \boldsymbol{r}_\rho^\top + \delta \boldsymbol{W}_{\mathrm{h}} \,. \tag{2}$$

If the geometry of the low-rank component is constant across simulations, i.e. the singular vectors $\boldsymbol{l}_\rho$ and $\boldsymbol{r}_\rho$ are the same up to a global transformation, then one should be able to isolate them by averaging over an ensemble of network initializations. To this end, we trained $K$ networks with identical data and task objectives, but different random initializations and minibatch updates. We expect that any differences in learned weights across networks should be attributable to a global transformation. Since the data and the task are identical across simulations, we expect that this transformation can be found by comparing the singular vectors of the $p \times N$ matrix $\boldsymbol{H}$, containing the hidden activity at the final time step upon readout, i.e. $H_i^\mu = h_{i,t=L}^\mu$.

For each experiment, denoted $k = 1 \ldots K$, we apply SVD to the corresponding activity matrix $\boldsymbol{H}^{(k)}$. We denote $\boldsymbol{U}^{(k)}$ and $\boldsymbol{V}^{(k)}$ the matrices whose columns are the left and right singular vectors, respectively, and $\boldsymbol{S}^{(k)}$ the diagonal matrix with the corresponding singular values, so that $\boldsymbol{H}^{(k)} = \boldsymbol{U}^{(k)} \boldsymbol{S}^{(k)} \boldsymbol{V}^{(k)^\top}$.

We find that while the singular value spectra $\boldsymbol{S}^{(k)}$ are highly consistent across runs, the singular vectors themselves are randomly orientated. We can use the right singular vectors matrices $\boldsymbol{V}^{(k)}$ as the orthonormal reference frames, with respect to which the hidden activity can be compared across simulations. In this rotated frame, the hidden activity is written as

$$\bar{\boldsymbol{h}}^{(k)} = \boldsymbol{R}^{(k)} \cdot \boldsymbol{h}^{(k)} \,, \tag{3}$$

where $\boldsymbol{R}^{(k)} = \boldsymbol{V}^{(k)^\top}$. Similarly, we can compare the learned weights across simulations via an analogous change of reference frame. In order for the currents $\boldsymbol{z}$ in Eq. 1 to transform in the same way as Eq. 3, i.e. $\bar{\boldsymbol{z}}^{(k)} = \boldsymbol{R}^{(k)} \cdot \boldsymbol{z}^{(k)}$, the recurrent weights must transform as

$$\bar{\boldsymbol{W}}_{\mathrm{h}}^{(k)} = \boldsymbol{R}^{(k)} \boldsymbol{W}_{\mathrm{h}}^{(k)} \boldsymbol{R}^{(k)^\top} \,. \tag{4a}$$

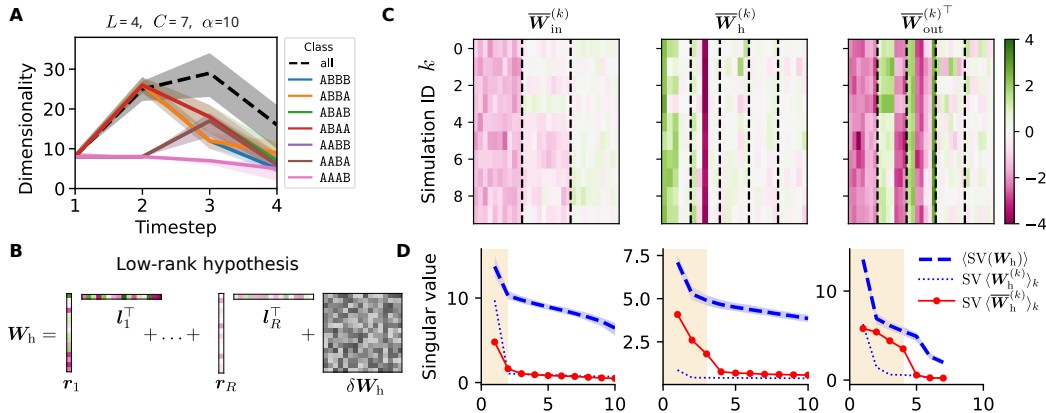

Figure 2: **Low-dimensional dynamics and low-rank connectivity in trained recurrent networks.** A. Dimensionality of hidden activity over time, measured as the number of singular vectors capturing 90% of the mean squared activity (medians +/- 5th-95th percentiles). Dimensionality is class-dependent; typically rising mid-sequence and decreasing toward the end. **B.** Recurrent connectivity matrix shown as a sum of a structured low-rank component (colored) and a random component (gray). **C.** Input, recurrent and output weight matrices from multiple simulations shown in a common rotated coordinate frame (see Eqs. 4). Each row corresponds to one simulation; matrices are flattened by concatenating rows (input/recurrent) or columns (output transpose). Only the dominant modes are shown (3 for input (left), 5 by 5 for recurrent (middle) and 5 for output (right) weights). Note the consistent structure across simulations. **D.** Low-rank structure extracted by averaging aligned weight matrices across simulations. Spectra of recurrent matrices are similar across simulations (dashed blue line: mean of SVs of the weight matrices from individual simulations +/- $5 - 95\%$ percentiles bands). Spectra of the average recurrent weights before and after change of basis (dotted blue vs red) differ only in their structured components (shaded red region, used to infer the rank of the structured part). Left: spectrum of input weights. Middle: recurrent where $R = 3$. Right: output.

and the input weights as

$$\bar{\boldsymbol{W}}_{\text{in}}^{(k)} = \boldsymbol{R}^{(k)} \boldsymbol{W}_{\text{in}}^{(k)} . \tag{4b}$$

Finally, in order for the outputs to remain unchanged, we need to transform the readout weights as

$$\bar{\boldsymbol{W}}_{\text{out}}^{(k)} = \boldsymbol{W}_{\text{out}}^{(k)} \boldsymbol{R}^{(k)\top} . \tag{4c}$$

In the case of a linear network, where $\phi$ is the identity, Eq. 1 are clearly invariant under rotations. If $\phi$ is the rectified-linear activation function, this is generally not the case. However, we see that the symmetry holds approximately, as shown by the fact that the entries of $\bar{\boldsymbol{W}}_{\text{h}}$ and $\bar{\boldsymbol{W}}_{\text{out}}$ are approximately the same across simulations (Fig. 2C). Our working hypothesis is that, modulo a simulation-specific rotation, differences in the recurrent weight matrices across simulations are accounted for by the "quenched" noise term $\delta \boldsymbol{W}_{\text{h}}$ in Eq. 2, while the low-rank part, supporting task demands, remains the same (Fig. 2D). Therefore, we can estimate the rank of the structured part by comparing the spectrum (singular values) of the average transformed weight matrices, $\langle \bar{\boldsymbol{W}}_{\text{h}} \rangle = K^{-1} \sum_k \bar{\boldsymbol{W}}^{(k)}$, with that of the average original weights matrix, $\langle \boldsymbol{W}_{\text{h}} \rangle = K^{-1} \sum_k \boldsymbol{W}^{(k)}$. While the former retains the common low-rank structure, in the latter, the random relative orientation of the structured part is averaged away. We estimate the rank $R$ of the common component to be the number of singular values of $\langle \bar{\boldsymbol{W}}_{\text{h}} \rangle$ larger than those of $\langle \boldsymbol{W}_{\text{h}} \rangle$. For the dataset presented here, $R = 3$, which corresponds to the number of singular values before the "elbow" in the spectrum of $\langle \bar{\boldsymbol{W}}_{\text{h}} \rangle$ (Fig. 2D).

In the next section, we explore how the structure of the weights, i.e. the dominant singular vectors, relate to the statistical and relational structure of the input sequences.

## 2.2 LOW-RANK RECURRENT WEIGHTS DRIVE RELATIONAL STRUCTURE IN THE LEARNED REPRESENTATIONS

The sequences used in this work are defined by temporal regularities described by a binary branching tree (Fig. 1A). A central question is how this relational structure is encoded in the RNN's recurrent connectivity. We first examine the cosine similarity of hidden activity across sequences (Fig. 3A). As the sequence unfolds, the similarity matrix progressively reflects *only* the transition type: sequences continued by a "same" transition (AA-type) become more similar to each other than to those continued by a "different" transition (AB-type), independent of token identity. This emerges clearly at $t = 2$, where the similarity matrix separates in two blocks corresponding to AA and AB. At later timepoints, these diagonal blocks strengthen, and additional off-diagonal blocks appear, reflecting higher similarity among sequences sharing the *most recent* transition. By the classification time ($t = 4$), however, the matrix *primarily* reflects the the *earliest* transitions: sequences sharing the initial branch of the tree exhibit more similar hidden activity than those diverging early.

Thus, the representations approximate a binary branching tree in which sequences are separated according to their transition structure in chronological order(Fig. 1A). To quantify the degree to which these representations exhibit hierarchical, tree-like geometry, we compute the *ultra-metric content* (UC) of the representations [20; 2] at each time step (Fig. 3B-C). UC measures how closely a set of representations conforms to an ultrametric (tree-like) space (Fig. 3B, see SI D. for details). We find that UC increases as the sequence unfolds (Fig. 3C, solid lines). The rise of UC beyond within-class baselines (dashed lines) reflects the increasing depth of the underlying tree. Importantly, this hierarchical structure is expressed in a compact, low-dimensional form: projecting the hidden activity onto the top singular vectors of the recurrent weight matrix $W_h$ recovers most of the ultrametric structure observed in the full representation (Fig. 3C). Finally, we note that the tree in Fig. 1A and 3B is not the only tree consistent with the representational geometry. One can construct a *backward* tree, starting from the last transition and branching in *reverse* order, which yields a different arrangement of the leaf classes. The cosine similarity matrix reflects contributions from *both* hierarchies: strong diagonal blocks correspond to early (forward) transitions, while the weaker off-diagonal blocks arise from shared *latest* transitions (backward hierarchy). Because the network processes sequences chronologically, the forward hierarchy is more salient, but the presence of structured off-diagonal similarity shows that the model retails information about late transitions as well. Thus, these deviations do not reflect noise; they reveal that the network's representations interpolate between the forward and backward relational structures inherent in the task.

Last, we asked whether the quality of the abstract representations, as measured by the UC score, correlate with the network's ability to generalize. For increasing number of classes, we computed both the mean UC score at the end of learning and the corresponding generalization accuracy. For networks of appropriate size, we observe a positive correlation: networks that settle into more strongly hierarchical representations generalize better, suggesting that the quality of the internal abstract configuration directly drives generalization (Fig. 3E)).

## 2.3 LOW-RANK NETWORK PERFORMANCE AND PERTURBATION EXPERIMENTS REVEAL FUNCTIONAL ROLE OF DOMINANT SINGULAR COMPONENTS

Next, we asked whether the low-rank component of the recurrent weights is sufficient to support the classification task, and what computational role the dominant singular components play in shaping abstract representations.

While averaging weight matrices across aligned networks reveals the approximate singular values $\{S_\rho\}$ of the low-rank part, the corresponding singular vectors cannot be cleanly extracted: as seen in Fig. 2D (middle panel), the low-rank structure and the noise components of $W_h$ have comparable magnitudes (as seen by comparing the dashed blue line with solid red line, and the gap between them), so the singular vectors of the full matrix do not reliably isolate the underlying low-rank subspace. Nevertheless, we can still test functional sufficiency by training explicitly rank-constrained networks.

We parametrized the recurrent weights matrix as a product $W_h = AB^\top$, where $A$ and $B$ are $\mathbb{R}^{N \times r}$ trainable matrices, and varied the maximum rank $r$. Networks of rank 3 (as well as 4 and 5) learn more slowly than full-rank models but ultimately match their memorization performance and achieve nearly identical generalization (Fig. 4A). In contrast, rank 1 and rank 2 networks show

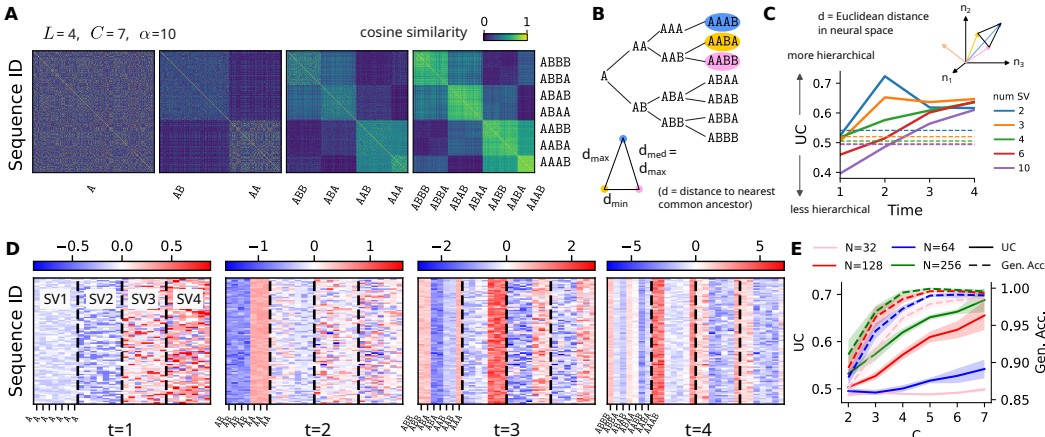

Figure 3: **Hierarchical structure of hidden representations. A.** Cosine similarity between hidden activities for all pairs of training sequences (test sequences show similar trends). As sequences unfold, the similarity matrix reveals a quasi-ultrametric organization reflecting the hierarchical structure of the sequences. **B.** Left: Hierarchical tree of abstract sequence classes. In an ultrametric space (bottom), distances between triplets of representations, as measured by their distance to their nearest common ancestor are restricted to equilateral or isosceles triangles with two long sides, disallowing isosceles with two short sides. A perfect hierarchical tree lies in an ultrametric space. **C.** Pairwise Euclidean distances between neural representations (top) yield ultrametric content (UC) values that approach–but deviate from–the ultrametric limit (bottom). UC over time for sequences of length $L = 4$. Solid lines: UC from projections onto singular vectors of the recurrent weights; four components capture the unfolding hierarchy. Dashed lines: UC within individual classes (baseline). **D.** Hidden activity projected onto the first four singular vectors of the recurrent connectivity. Rows: individual sequences; inner columns: class membership; outer columns (separated by dashed lines): the first four singular vectors. **E.** UC scores (left axis) and generalization accuracy (right axis) as a function of number of abstract classes. Higher UC scores correlates with higher generalization performance.

marked deficits; their confusion patterns reflect their inability to distinguish abstract classes that are close in representational space (Fig. 4B-C).

To interpret the functional role of the singular components, we performed *singular vector ablation* on trained networks with recurrent weights $\boldsymbol{W}_{\mathrm{h}}$. After training a rank-constrained network, we removed selected singular vectors from the recurrent weights, then simulated the network on all sequences, and computed its performance. The perturbed recurrent weight matrix where the $a$-th singular component has been removed would be $\tilde{\boldsymbol{W}}_{\mathrm{h}}^{(a)} = \boldsymbol{W}_{\mathrm{h}} - s_a \, \boldsymbol{l}_a \, \boldsymbol{r}_a^{\top}$, such that that $\tilde{\boldsymbol{W}}_{\mathrm{h}}^{(a)} \boldsymbol{r}_a = \boldsymbol{0}$ i.e. any activity along $\boldsymbol{r}_a$ is not propagated to the next step.

Examining hidden activity projected onto the dominant singular vector reveals its functional meaning. At $t = 2$, projections along the first singular vector split cleanly into two blocks reflecting same/different transitions (`AA`/`BB` vs. `AB`/`BA`, Fig. S4). At subsequent timesteps, each block subdivides according to the *most recent* transition. By $t = 4$, this yields perfect decoding of abstract classes: the projection carries integrated information about the full transition history, hence the hierarchical generative tree (Fig. 4D left and Fig. S4B). However, when the dominant singular component is removed, this integrated structure collapses. The projection onto $\boldsymbol{r}_1$ now reflects *only the very last transition*, and information about earlier transitions is lost (Fig. 4D, right and Fig. S4B). Therefore, the projected activity can only discern between sequences of type `**AB` vs `**AA`, but not *within* these groups. Correspondingly, the cosine similarity matrix loses its hierarchical block structure (Fig. 4C), and the confusion matrix shows that sequences are grouped solely by their *most recent* transition (Fig. 4B). Removing the second singular component produces only mild degradation, indicating that the dominant singular mode plays a special role in integrating sequential transition information over time.

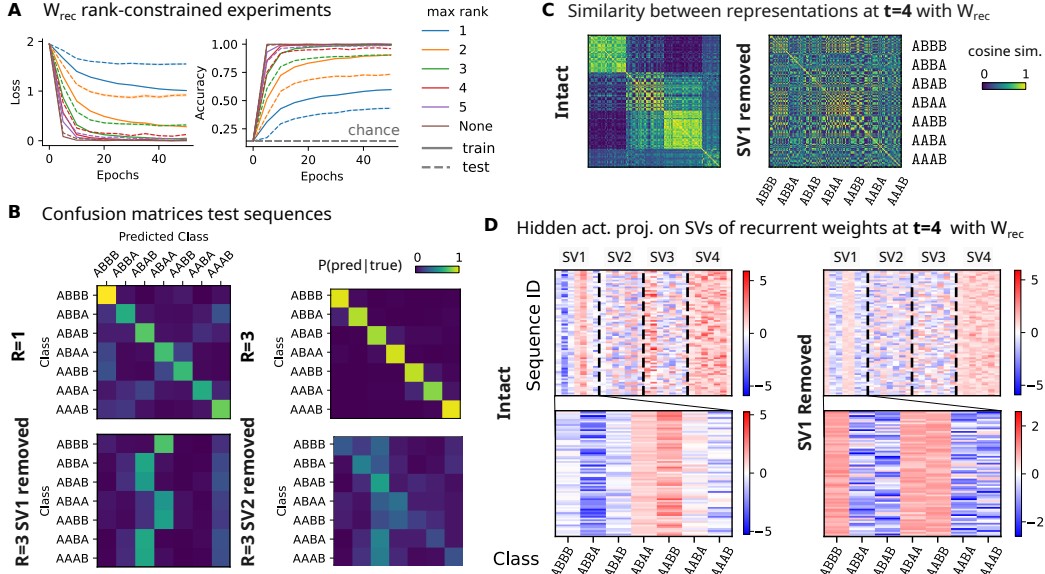

Figure 4: **Perturbation of singular components reveals how low-rank dynamics support abstraction. A.** Low-rank recurrent connectivity is sufficient for the task. **B.** Confusion matrices reveal the nature of errors. Perturbing a trained rank-3 recurrent matrix by removing individual singular components (bottom) show that eliminating the leading component (SV1) causes sequences to be classified almost solely by their last transition, "same" vs "different" (i.e. AA vs AB, collapsing many classes (bottom left). Removing SV2 yeilds milder distortions (bottom right). **C.** Cosine similarity matrices show that the structured clustering present in the intact network is disrupted when SV1 is removed. **D.** Projections of hidden activity onto the dominant (right) singular vectors illustrate the perturbation's effect: with SV1 removed, the projection along that direction reflects only the most recent transition, while other projections carry little class information. In the intact network, SV1 (and to a lesser extend SV2-SV3) integrate transition information over time, supporting the formation of hierarchical, class-specific representations (see Fig. S4 for full time courses).

More generally, $r_1$ retains a consistent functional meaning across the entire sequence: at every timestep, the projection of hidden activity onto $r_1$ selectively encodes the most recent same/different transition (Fig. S4). Removing the dominant singular component prevents this transition information from being propagated forward in time, so the network can no longer accumulate a history of transitions. As a result, the hierarchical, tree-like representation built from integrating successive transitions is disrupted, and only the most recent transition is retained.

Finally, in full-rank networks, ablating the dominant component produces qualitatively similar effects (Fig.,S4), though somewhat attenuated due to residual noise components. The perturbed network still represents the most recent transition, but this information fails to propagate through time, preventing accumulation of relational structure.

Together, these findings demonstrate that the leading singular components of the recurrent weight matrix implement temporal integration of relational ("same/different") transitions, enabling the network to construct tree-like representations of abstract sequence structure.

## 3 LOW-RANK CLASSIFICATION SCAFFOLD LEADS TO BETTER GENERALIZATION IN PREDICTION TASKS

We next asked whether low-rank recurrent structure that emerges in the classification model is task-dependent. To test this, we trained the same architecture on a next-token prediction task, where the network must output the next item at each time step. Although the classification and prediction differ in their learning objectives–global sequence classification versus stepwise forecasting–they

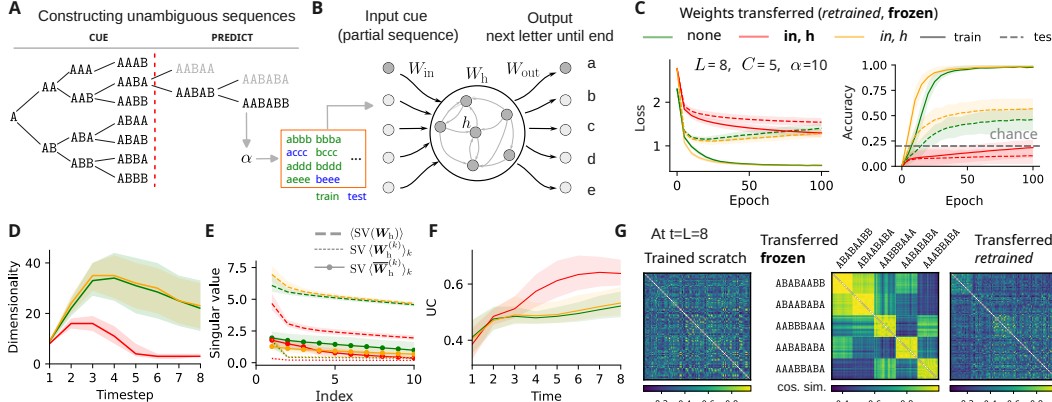

Figure 5: **Transfer from classification improves prediction learning. A.** To construct unambiguous sequences for the prediction task, we apply extreme pruning to a binary tree (light gray), retaining only one path per cue (black). The red dashed line marks the boundary between the cue (input) and the continuation to be predicted. Unique tokens are then replaced by letters drawn from an alphabet of size $\alpha$. **B.** Left: a standard RNN is trained from scratch to predict each next token in the continuation sequence. Right: prediction networks are either initialized with weights transferred from a classification-trained RNN and then retrained (italic labels), or evaluated with transferred recurrent weights kept fixed (bold). **C.** Loss (left) and accuracy (right) for prediction networks under three initializations: random (green), transferred recurrent weights kept frozen (red; only output weights trained), and transferred recurrent weights used as initialization and then jointly trained (orange). Means and standard deviations shown across simulations. Only the latter condition accelerates memorization and improves generalization. To reduce overfitting, training was stopped at epoch 25 in all cases. **D.** Dimensionality of hidden activity over time, measured as the number of singular vectors capturing $90\%$ of the mean squared activity (Median +/- $5 - 95\%$ percentile bands). Dimensionality decreases mainly when transferred recurrent weights are frozen (red); networks trained from scratch (green) or fine-tuned from classification (orange) maintain higher dimensionality. **E.** Spectra of recurrent matrices (averaged over simulations, dashed), and spectra of their averages before and after change of basis (dotted vs solid). Only frozen transferred weights (red) yield a clear structured low-rank component ($R = 3$). Prediction networks trained from scratch (green) show no low-rank structure; those initialized from classification (orange) also lack clear low-rank structure, though their spectrum is less flat than randomly initialized networks. **F.** Ultrametric content (UC) of hidden representations over time for each initialization condition (as in Fig. 3C, mean UC +/- $5 - 95\%$ percentile bands). **G.** Cosine similarity matrices between hidden representations for each initialization condition. Left: prediction net trained from scratch. Middle: weights transferred from classification to prediction and frozen. Right: weights transferred from classification and then retrained. Only the transfer-and-freeze condition preserves the hierarchical, low-dimensional representations.

share key computational demands: encoding time-varying inputs, maintaining memory across timesteps, and extracting the latent relational structure underlying the sequences.

A key challenge for training the prediction network was constructing unambiguous sequences. In classification, the network receives full sequences; in prediction, the network is cued with a partial sequence and must recursively generate the continuation, feeding each predicted item back as input. Using the same sequences as in classification proved problematic: many cues were ambiguous, producing multiple valid continuations, and the network defaulted to memorizing training examples rather than generalizing. To resolve this, we generated cues from the binary tree structure and then pruned the tree, retaining only a single valid continuation for each cue (Fig. 5A). This ensured that each prediction target was uniquely defined.

Using this protocol, prediction networks trained on the same pruned sequence distribution as classification networks (Fig. 5B) maintained strong memory capacity (Fig. S2, top) but generalized more poorly (Fig. S2, bottom). Their internal dynamics also differed markedly: in contrast to classification networks, dimensionality remained high throughout the sequence (Fig. 5D, green), the recurrent

weights lacked low-rank structure (Fig. 5E), and their representations showed weaker hierarchical organization (Fig. 5F). Thus, low-rank dynamics do not arise merely from processing the same input statistics; they appear specific to tasks requiring global integration across the full sequence [21; 10].

Next, we examined whether transferring the weights from a classification-trained model could accelerate convergence and improve generalization in the prediction task. We transferred recurrent and input weights (and biases) from a classification-trained model and tested several regimes: freezing the transferred weights (except outputs) (Fig. 5E, red), or using them as initialization for continued training (Fig. 5E, orange). Freezing the weights offered no benefit over chance (Fig. 5E). In contrast, jointly initializing both input and recurrent weights from the classification task led to faster learning of the test set and improved asymptotic generalization performance (Fig. 5E). We find that this benefit is mostly due to transferring the recurrent weights and not the input weights (Fig. S3A, compare purple with cyan).

Finally, to test whether these improvements are specifically due to the abstract scaffold and not generic pretraining, we trained an autoencoder model to reconstruct sequences (Fig. S3E). Transferring encoder weights to the prediction model sped up memorization but did not improve generalization, unlike classification pretraining. This dissociation indicates that the transfer benefit arises from the learned abstract relational scaffold, not from shared data statistics or generic pretraining.

## 4 CONCLUSION AND DISCUSSION

Our findings demonstrate that abstract temporal structure can emerge spontaneously in recurrent neural networks trained on sequence-level labels. Without architectural biases or intermediate supervision, vanilla RNNs learned compact, low-dimensional internal representations that reflected the temporal hierarchical structure of the input sequences and generalized across different token instantiations. This abstraction was supported by low-rank recurrent connectivity, which shaped hidden-state trajectories into a shared low-dimensional manifold aligned with the binary branching structure of the data.

By explicitly constraining network rank and perturbing individual singular components, we identified a causal role for the dominant recurrent mode: it integrates past and current same/different transitions at each timestep, enabling the progressive separation of classes along the generative tree as the sequence unfolds. Removing this component erases the memory of earlier transitions and collapses tree-like structure, demonstrating the mechanistic function of the recurrent architecture's leading singular direction.

Crucially, we showed that this learned low-rank scaffold is reusable: transferring recurrent weights from classification to a next-token prediction task improves generalization, whereas generic pretraining (e.g. autoencoder) does not. These findings suggest that even simple recurrent architectures can internalize latent statistical structure in a flexible, and reusable way, offering a potential computational mechanism for abstraction in biological circuits. Moreover, the geometry and dimensionality of these dynamics provide concrete predictions for neural population activity. In ongoing work, we are applying these insights to electrophysiological recordings from rodents learning structurally analogous sequences, testing whether hippocampal and frontal populations exhibit similar low-dimensional, transition-integrating dynamics. Future work could extend this framework to more complex, naturalistic sequence statistics, investigate the developmental trajectory of abstraction during learning, or explore how these low-dimensional structures support compositional reasoning and flexible behavior.

Several limitations remain. First, we focus on algebraic patterns generated by deterministic binary trees and fixed sequence lengths. Whether similar low-rank structure and transferable representations arise for probabilistic grammars, naturalistic statistics, or variable-length sequences remains an open question. Second, real-world environments contain noise, stochastic transitions, and hierarchical dependencies at multiple timescales; these may reshape how abstract representations form. Third, Third, our models were trained on one objective at a time, whereas biological learners solve multiple tasks simultaneously. Finally, we analyzed only end-of-training representations; understanding how abstraction develops over learning [17] is an important direction for future work.

REPRODUCIBILITY STATEMENT

We affirm that this manuscript provides sufficient detail about the architecture, training procedure, and sequence generation process to allow reproduction of the main experiments. Specific parameter choices (e.g., sequence length, number of classes, alphabet size, initialization regimes) are described and the entire parameter set required to reproduce figures are reported in the Supplementary Material. The sequences used in the paper are synthetically generated using a deterministic procedure described in detail in the methods section and supplementary material. The main code (sequence generation and model training) are available at the following link: `https://anonymous.4open.science/r/ICLR2026_submission-2F3D/`. Upon publication, we will also release the code for the analyses, together with documentation and instructions for reproducing the experiments.

ETHICS STATEMENT

We affirm that this work has been conducted in accordance with the ICLR Code of Ethics. In particular, we uphold the principles of responsible stewardship, high standards of scientific excellence, honesty and transparency, fairness, respect for privacy and confidentiality, and a commitment to minimizing harm while contributing positively to society and human well-being. All data used in the study are synthetically generated and do not represent real-world individuals or environments. This work is focused on understanding how abstract structure can emerge in neural networks through carefully controlled simulations. As such, it poses no foreseeable safety, security or privacy risks. The models used are standard RNNs trained on sequences of synthetic tokens and are not applicable to domains where deception, discrimination, impersonation, or surveillance could arise. There is no direct or indirect link to systems that could be weaponized or otherwise cause harm. In terms of environmental impact, the computational footprint of the experiments is relatively very low. The models are lightweight, training times are short, and experiments were conducted on a modest compute budget without use of large-scale infrastructure. The study does not promote unsustainable resource usage or fossil fuel dependence.

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

# SUPPLEMENTARY INFORMATION

## A. NEURAL NETWORK MODELS

**Classification model**    In this task, the output of the RNN model is the probability distribution $\boldsymbol{p}$ over $C$ classes, therefore in the $(C-1)$-dimensional simplex $\Delta^{C-1}$. We choose to minimize a cross-entropy (CE) function

$$l(X, y) = -\sum_{c=1}^{C} y^c \log p^c(X) , \tag{S1}$$

where $y^c$ is the target probability that the sequence $X$ belongs to class $c$ (1 if $c$ equals the true class, 0 otherwise). The output is computed after the $L$-th (last) element in the sequence $X$:

$$\boldsymbol{p} = \mathrm{softmax}\big(\boldsymbol{W}_{\mathrm{out}} \cdot \boldsymbol{h}_L + \boldsymbol{b}_{\mathrm{out}}\big) \tag{S2}$$

where $\boldsymbol{W}_{\mathrm{out}} \in \mathbb{R}^{C \times N}$ is the output weight matrix, i.e.

$$p^c = \frac{\mathrm{e}^{\boldsymbol{w}_{\mathrm{out}}^c \cdot \boldsymbol{h}_L + b_{\mathrm{out}}^c}}{\sum_j \mathrm{e}^{\boldsymbol{w}_{\mathrm{out}}^j \cdot \boldsymbol{h}_L + b_{\mathrm{out}}^j}} . \tag{S3}$$

where $\boldsymbol{w}_{\mathrm{out}}^c$ and $b_{\mathrm{out}}^c$ are the $c$-th row of $\boldsymbol{W}_{\mathrm{out}}$ and the $c$-th component of $\boldsymbol{b}_{\mathrm{out}}$, respectively.

**Prediction model**    In the prediction task, the output is a probability distribution over the letter in the alphabet, $\boldsymbol{p} \in \Delta^{\alpha-1}$. Here the objective is to predict the next letter in the sequence. Given a starting cue letter, the network outputs the next letter, and this next letter then serves as a cue for the prediction of the next letter. Therefore the loss function that we minimize is the cross entropy between sequence elements at time $t$ and those at time $t+1$.

$$l(\boldsymbol{X}, \boldsymbol{Y}) = -\sum_{t=1}^{L-1} \sum_{i=1}^{\alpha} Y_{t+1}^i \log p_t^i , \tag{S4}$$

where $Y_{t+1}^i$ is the next letter in the sequence, and $p_t^c$ is probability distribution of the next letter as predicted by the network, given by

$$\boldsymbol{p}_t = \mathrm{softmax}\big(\boldsymbol{W}_{\mathrm{out}} \cdot \boldsymbol{h}_t + \boldsymbol{b}_{\mathrm{out}}\big) . \tag{S5}$$

where $\boldsymbol{W}_{\mathrm{out}} \in \mathbb{R}^{\alpha \times N}$.

**Reconstruction model**    For this task, we consider an autoencoder architecture composed of an RNN encoder and an RNN decoder, with the same number of recurrent units, $N$.
At the end of the input sequence, a fully connected linear layer maps the recurrent activity of the encoder to an $n$-dimensional latent space.
The latent activity is then fed as a constant input to the decoder RNN through a linear. All the weights in the network are updated via batch gradient descent through BPTT.

$$l(\boldsymbol{X}, \tilde{\boldsymbol{X}}) = -\sum_{t=1}^{L-1} \sum_{i=1}^{\alpha} \boldsymbol{x}_t^i \log \tilde{\boldsymbol{x}}_t^i , \tag{S6}$$

where $\boldsymbol{x}_t^i$ is the input sequence (and the target), $\tilde{\boldsymbol{x}}_t^i$ is the reconstructed sequence, given by

$$\tilde{\boldsymbol{x}}_t^i = \mathrm{softmax}\big(\tilde{\boldsymbol{W}}_{\mathrm{out}} \cdot \tilde{h}_t + \tilde{\boldsymbol{b}}_{\mathrm{out}}\big) \tag{S7}$$

$$\tilde{h}_t = \phi\big(\tilde{\boldsymbol{W}}_{\mathrm{h}} \cdot \tilde{h}_{t-1} + \tilde{\boldsymbol{W}}_{\mathrm{lat}} \cdot \boldsymbol{z} + \tilde{\boldsymbol{b}}_{\mathrm{h}}\big) \tag{S8}$$

$$\tilde{\boldsymbol{z}} = \boldsymbol{W}_{\mathrm{lat}} \cdot \boldsymbol{h}_L + \boldsymbol{b}_{\mathrm{lat}} \tag{S9}$$

$$h_t = \phi\big(\boldsymbol{W}_{\mathrm{h}} \cdot h_{t-1} + \boldsymbol{W}_{\mathrm{lat}} \cdot \boldsymbol{x}_t + \boldsymbol{b}_{\mathrm{h}}\big) \tag{S10}$$

with $\boldsymbol{h}_0 = \boldsymbol{0}$ and $\tilde{\boldsymbol{h}}_0 = \boldsymbol{0}$ and where $\boldsymbol{h}, \tilde{\boldsymbol{h}}, \boldsymbol{b}_{\mathrm{h}}, \tilde{\boldsymbol{b}}_{\mathrm{h}} \in \mathbb{R}^N$, $\boldsymbol{x}_t, \tilde{\boldsymbol{x}}_t, \tilde{\boldsymbol{b}}_{\mathrm{out}} \in \mathbb{R}^\alpha$, $\tilde{\boldsymbol{W}}_{\mathrm{out}} \in \mathbb{R}^{\alpha \times N}$, $\tilde{\boldsymbol{W}}_{\mathrm{lat}} \in \mathbb{R}^{N \times n}$, $\boldsymbol{b}_{\mathrm{lat}}, \tilde{\boldsymbol{z}} \in \mathbb{R}^n$, and $\boldsymbol{W}_{\mathrm{lat}} \in \mathbb{R}^{n \times N}$.

## B. Generating Structured Sequences

We construct sequences designed to exhibit specific regularities. Two key parameters define the sequence space: the sequence length $L$, and the number of distinct letters $m$ used in each sequence, with $m < L$, to ensure repetition and thus induce "regularity" in the sequence. For example, consider $L = 2$ and $m = 2$: the only possible sequence is $AB$. With $L = 3$ and $m = 2$, possible patterns include $AAB$, $ABB$ and $ABA$, each exhibiting a different structure of repetition and alternation–e.g. "same different" vs "different same". These patterns define what we refer to as *algebraic patterns* [11], or abstract temporal schemas. The number of such abstract patterns correspond to the number of ways to partition a sequence of $L$ items into groups of at least one element is given by

$$\sum_{n_1,n_2,\ldots,n_m} \frac{L!}{n_1!n_2!\cdot\ldots\cdot n_m!} \,, \tag{S11}$$

where the sum over $n_1, ..., n_m$ is over partitions of $L$, i.e. they satisfy the constraint $\sum_{i=1}^{m} n_i = L$.

To instantiate each abstract pattern with concrete sequences, or tokens, we select $m$ distinct letters from a given alphabet of size $\alpha$. The number of such choices is

$$C(\alpha, m) = \frac{\alpha!}{(\alpha - m)!m!} \,. \tag{S12}$$

Thus, we produce $p$ instances of the same sequence type, given by the multiplication of the two expressions above

$$p = \frac{\alpha!}{(\alpha - m)!m!} \sum_{n_1,n_2,\ldots,n_m} \frac{L!}{n_1!n_2!n_m!} \tag{S13}$$

These sequences constitute a fraction of the total number of possible sequence configurations of length $L$ given by $\alpha^L$. We use these structured sequences to train our model. In this manuscript, we used $80\%$ of the sequences (chosen randomly) to train the models, and the remaining $20\%$ to test the models. Choosing sequences randomly meant that networks were highly likely to have seen all the letters used in the data, but never in the configurations present in the test data.

**Sequences produced using a binary branching tree**   In the special case of $m = 2$, this generative procedure defines a binary branching tree: each additional token in the sequence corresponds to a binary split (of an existing node), producing a hierarchical structure over the abstract classes (Fig. 1A, main text). Throughout the main text, we focus on this case to study how networks learn and represent such tree-structured temporal regularities.

**Generating unambiguous sequences for testing generalization in prediction networks**   As mentioned in the main text, sequences generated from terminal nodes of the full binary branching tree naturally contain overlapping elements across classes. This overlap poses a challenge for evaluating generalization in the prediction task, which requires prompting the network with a partial sequence (of length $L_{cue} < L$), and having it recursively predict the remaining $L_{retr} = L - L_{cue}$. This meant that a partial cue could theoretically lead to multiple different retrieval options, leading to ambiguous sequences. When ambiguous cues were used, the prediction network often defaulted to generating familiar training-set continuations, and was unlikely to retrieve sequences in the test set. This made it difficult to assess its ability to generalize to unseen sequences. To address this, we designed a procedure to construct unambiguous classes for testing prediction performance. Specifically, we generated cues of length $L_{cue}$ using the full tree structure, and then appended unique combinations of $m$ letters beyond that point, effectively pruning all but one continuation path from the node $L_{cue}$ onwards. Conceptually, this amounts to constructing a tree of depth $L$, from which all-but-one branches from node $L_{cue}$ to the terminal nodes $L$ of the tree are pruned (Fig. 4A, main text). This procedure meant that for a specific choice of $L_{cue}$ and $L$, many different combinations of classes are possible, from which we sample $C$ at a time (Fig. 4A of main text, black sequences sampled, gray sequences not sampled). To make sure that any effect that we observe is not due to a specific choice of this "class combination ($CC$)", we run many different simulations with different samplings, with a maximum of $CC = 20$ unique class combinations, when available.

## C. PHASE DIAGRAMS

To generate the phase diagrams for both the classification network (Fig. S1) and the prediction network (Fig. S2) where we vary the number of classes $C$ and the length of the sequences $L$, we used the pruning procedure to construct unambiguous sequences. This also allowed us to run transfer experiments (Fig. S3).

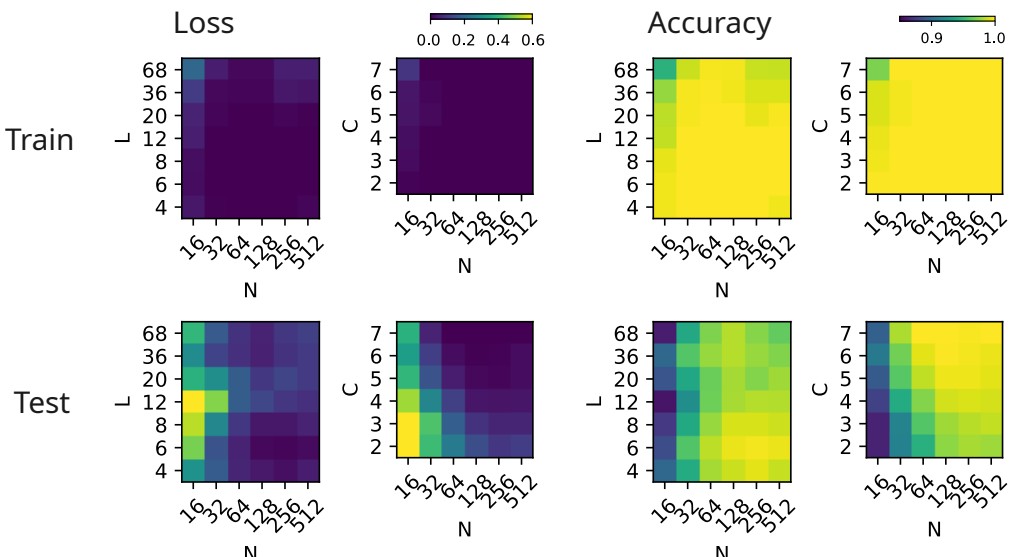

Figure S1: **Classification Task**. Phase diagrams showing train/test loss and accuracy as functions of network size $N$ (x-axis) and either sequence length $L$ or number of classes $C$ (y-axes). When $L$ is varied, the number of classes is fixed to $C = 4$. When $C$ is varied, the sequence length is fixed to $L = 7$.

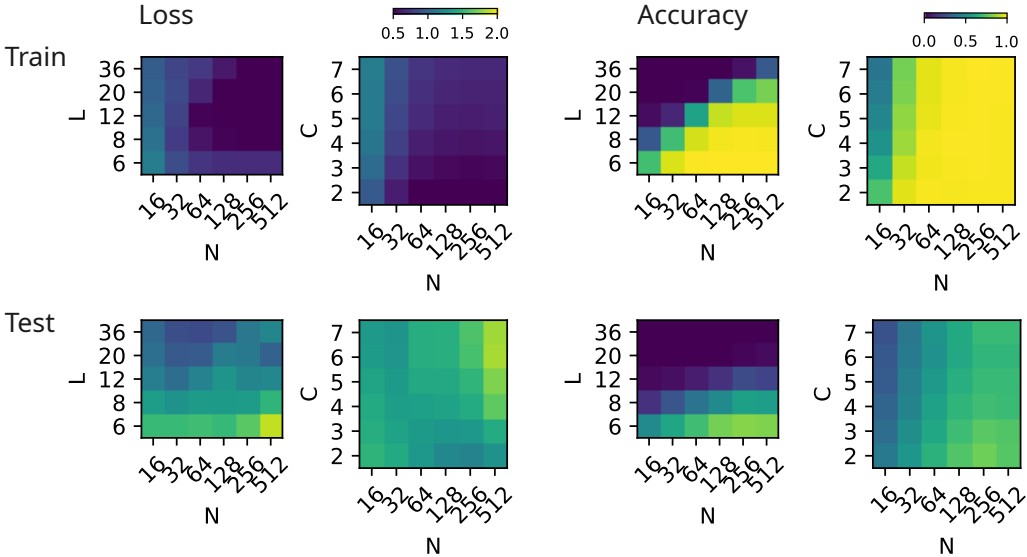

Figure S2: **Prediction Task**. Phase diagrams showing train/test loss and accuracy as functions of network size $N$ (x-axis) and either sequence length $L$ or number of classes $C$ (y-axes). Networks reliably memorize training sequences, but show poor generalization to held-out sequences. Longer sequences and more classes require larger networks.

## D.  ULTRAMETRIC CONTENT

Given a set of $p$ representations $\{\boldsymbol{h}^\mu\}_{\mu=1}^p$, we first compute the distance between any of the pairs $\boldsymbol{h}^\mu, \boldsymbol{h}^\nu$. Fig. 3B of main text shows a graphical depiction of sequences constructed using the binary branching process. Within this tree, the distance between any two terminal nodes (shown in color) can be defined as the distance to the nearest common ancestor. In order to compute the distance between pairs of real-valued neural representations, we simply compute the Euclidean distance between vectors of neural activity at any time-point (Fig. 3C, main text, top). We then take triplets of representations, $(\boldsymbol{h}^\mu, \boldsymbol{h}^\nu, \boldsymbol{h}^\rho)$. Denoting by $d_{\min}^{\mu\nu\rho}$ the edge of minimal length in the corresponding triangle, $d_{\max}^{\mu\nu\rho}$ the edge of maximal length and $d_{\text{med}}^{\mu\nu\rho}$ the edge of intermediate length, the ultrametric content [20] is defined as

$$\text{UC} = \frac{1}{N_{\text{tri}}} \sum_{1 \leq \mu < \nu < \rho \leq p} \frac{\log \delta_{min}^{\mu\nu\rho} - \log \delta_{med}^{\mu\nu\rho}}{\log \delta_{min}^{\mu\nu\rho} + \log \delta_{med}^{\mu\nu\rho}}, \tag{S14}$$

where $\delta_{min} = d_{\min}/d_{\max}$ and $\delta_{med} = d_{\text{med}}/d_{\max}$.

In Fig. 3C of main text, we compute the UC for the set of representations corresponding to all sequences (hidden activity at any given time-point, solid lines). As a control, to verify that the tree structure emerges due to the configuration of classes relative to one another, we also compute the UC at the end of the sequences, restricted to triplets belonging to the same class. It can be seen that towards the end of the sequence, as the class membership becomes apparent, the UC computed over all representations is above the UC values computed for triplets restricted to individual classes (0.5, Fig. 3C, main text).

If we plot $\delta_{min} = f(\delta_{med})$, triplets that satisfy the triangular inequality lie above the line $\delta_{min} = 1 - \delta_{med}$, while triplets that satisfy the ultra-metric inequality lie on the vertical line where $\delta_{med} = 1$. Triplets that are equilateral triangles lie at the point $\delta_{min} = \delta_{med} = 1$.

Eq. S14 gives a measure of the overall closeness of the cloud of triplets to the fully ultra-metric limit. This quantity ranges from 0 (all triplets forming isosceles triangles with two short sides) to 1 (for a fully ultra-metric set: equilateral triangles and isosceles triangles with two long sides).

## E.  TRANSFER EXPERIMENTS

We studied three tasks using an RNN: one trained to classify sequences based on the abstract temporal structure of said sequences, a second trained to predict the next item in a sequence, and a third trained to reconstruct sequences.

**Transferring classification pretrained weights to prediction network.**   While both classification and prediction trained networks exhibit good memory capacity as shown by their performance on sequences used in the training set, only the classification-trained network generalizes well across a wide range of parameters, particularly for longer sequences (Figs. S1 and S2). We found that the classification network achieves this via a progressive dimensionality reduction in its dynamics as the sequence unfolds (Fig. 2A, main text), which we linked to a low-rank structure in the recurrent connectivity matrix. In contrast, despite relying on similar sequence statistics (i.e. temporal patterns present in the classes), the prediction-trained network does not exhibit low-rank recurrent structure (Fig. 5D, main text). We note that one-step transition probabilities are not enough to perform the prediction task, and at any given time, the network requires knowledge of both the class and letter to perform this task. We hypothesized that the abstract structure learned by the classification network could be leveraged to improve prediction. To test this, we implemented multiple transfer learning protocols: either 1) initializing the prediction network's weights and biases from a trained classifier and then freezing them (i.e. no more learning), or 2) initializing from the classifier and training afterwards. We applied these protocols separately to the input weights, recurrent weights, or both (Figs. S3A). When all parameters were frozen (protocol 1, red curve), prediction performance remained at chance (Fig. S3A). Freezing only the recurrent weights (blue curve) allowed for learning of memorization but not generalization. Freezing only input weights (pink) permitted learning but generalization was slow relative to learning from scratch (green).

Interestingly, when recurrent and input weights were transferred separately following protocol 2 (cyan and purple), we observed modest improvements in both memorization and generalization

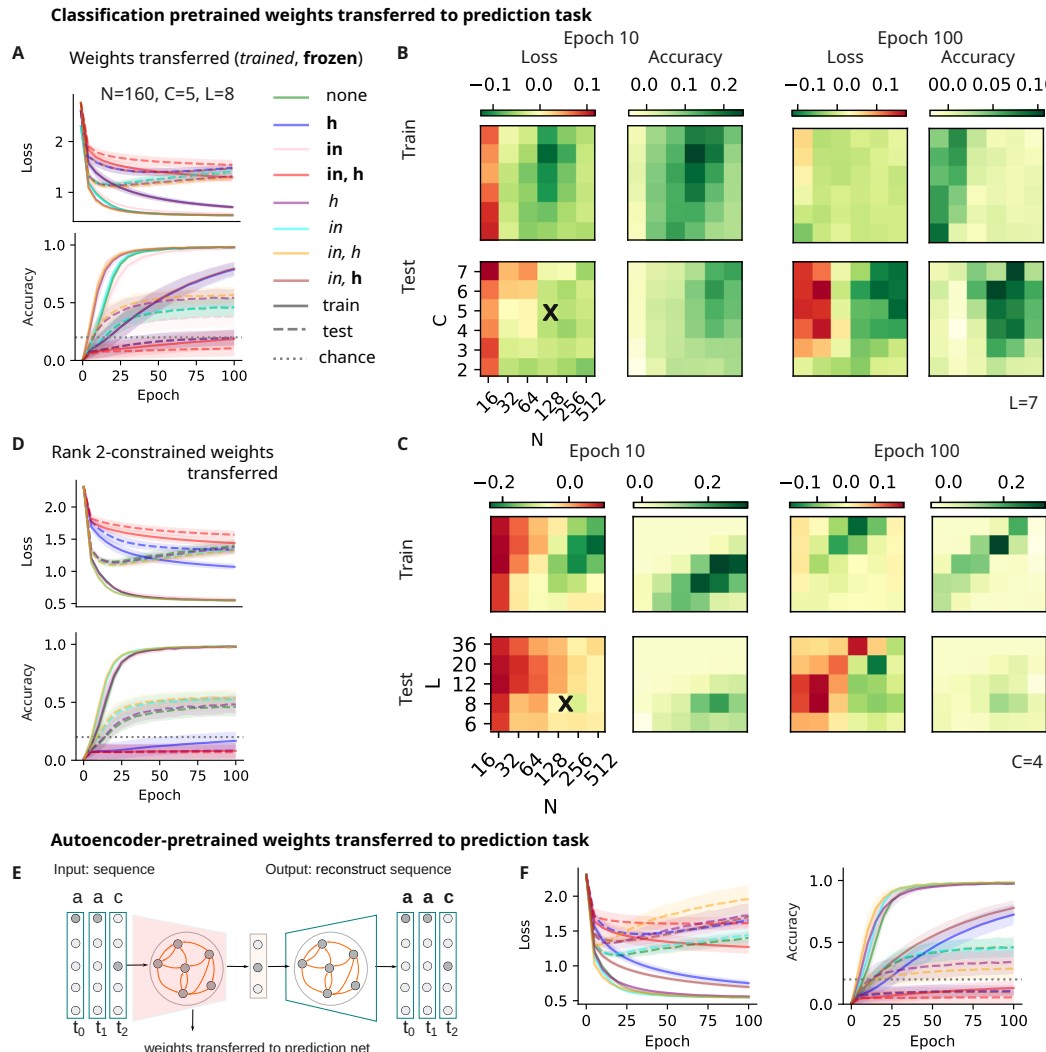

Figure S3: **The effect of classification and reconstruction pretraining on prediction tasks. A.** Loss (top) and accuracy (bottom) of prediction networks under three different initialization protocols: random initialization in the lazy regime (green), parameters from the classification pretraining transferred and kept frozen (blue, pink and red), and continued training (purple, cyan and orange). Finally, brown indicates frozen recurrent weights and retrained input weights. Lines indicate the mean across simulations; shaded areas show standard deviations. **B.** Difference in loss and accuracy between transfer-initialized and randomly initialized prediction networks, as a function of network size $N$ (x-axis) and number of classes $C$ (y-axis), for $L = 7$. These phase diagrams are plotted for early training (epoch 10) and late training (epoch 100). Green areas indicate improved performance through transfer. The black cross marks parameter configuration shown in panel **A**. **C.** Same as **B** but with sequence length $L$ on the y-axis, for fixed number of classes $C = 4$. **D.** We pretrained rank-constrained networks on the classification task and then transfer the weights onto the prediction network following the same protocol as in **A**. **E.** We perform the same transfer experiments by pre-training on a reconstruction task, with the RNN as the encoder of an auto-encoder architecture. **F.** The loss and activity learning curves show that transfer makes it faster to learn the training data, but it does not lead to an improvement in the generalization performance.

(Fig. S3A). However, combining both (input + recurrent; orange) led to sizeable gains in learning speed and generalization. To evaluate the robustness of this transfer, we ran this transfer protocol across a wide range of parameters, recurrent layer size $N$, number of classes $C$ and sequence length $L$ (Figs. S3B and C). Comparing against networks trained from scratch, we found that the transfer

protocol 2 (input + recurrent) improved performance (in both memorization and generalization) early (epoch 10) and late (epoch 100) in training (green areas) for large-enough networks, but with no to negative effects in the performance of very small networks (red areas).

**Transferring reconstruction pretrained weights to prediction network.** To control that any improvement in generalization performance when transferring classification-pretrained weights to the prediction network is due to the abstract scaffold and not due to a "warm start" hypothesis, we trained a an autoencoder model. We then transferred its encoder weights following exactly the same protocol as before. We observe in this case that although the network learns faster in almost all the protocols in which weights are transferred, generalization does not improve, as in the classification-pretrained case. We conclude from this that faster learning can be attributed to the warm start provided by pretraining the network, but for better generalization, only the abstract scaffold results in transfer.

## F. PERTURBATION EXPERIMENTS

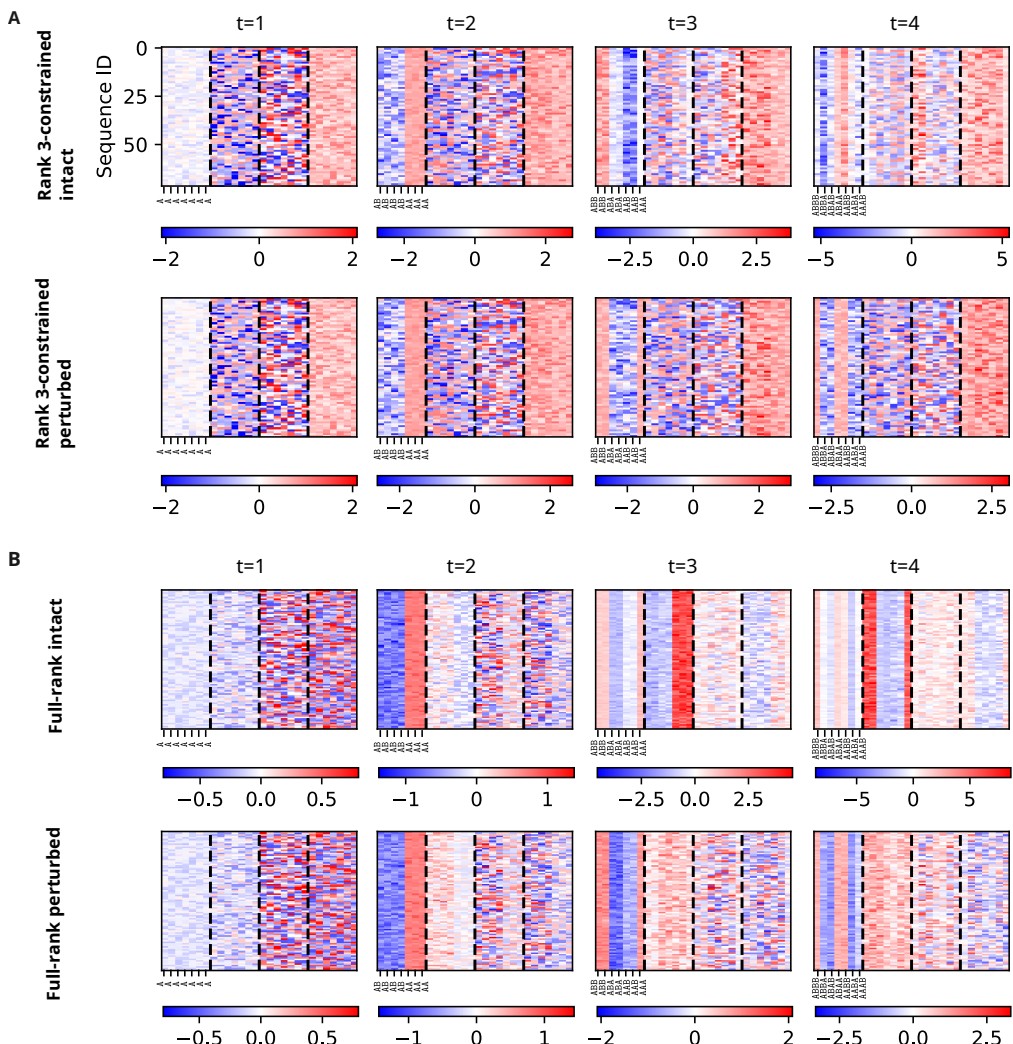

Figure S4: Perturbation experiments highlight the role of the singular components of the structured part of the recurrent connectivity. The RNN with a rank-3 recurrent weight matrix performs the task with high accuracy, approaching the performance of a full-rank one. In both cases, can see that the projections along the first principal components (**A**, top and **B**, top). By simulating a network where the dominant singular vector is removed from the recurrent weight matrix, we gain an insight into the role of the low-rank structured part (**A**, bottom and **B**, bottom). In both cases we notice the following. *i)* In the intact network the projections along the first singular components jointly carry information about the distinct classes into which the structure of sequences (or partial, processed so far) can be categorized. *ii)* When the dominant singular component is removed, the only projection that obviously carries information about the structure is SV1 itself: in particular, it is only informative of the latest seen transition ("same" vs "different"); since SV1 is removed from the recurrent weights, this is not propagated to the next time step, preventing the network from memorizing the whole structure.

## G. CLASSIFICATION TASK RESULTS WITH THE "RICH" INITIALIZATION REGIME

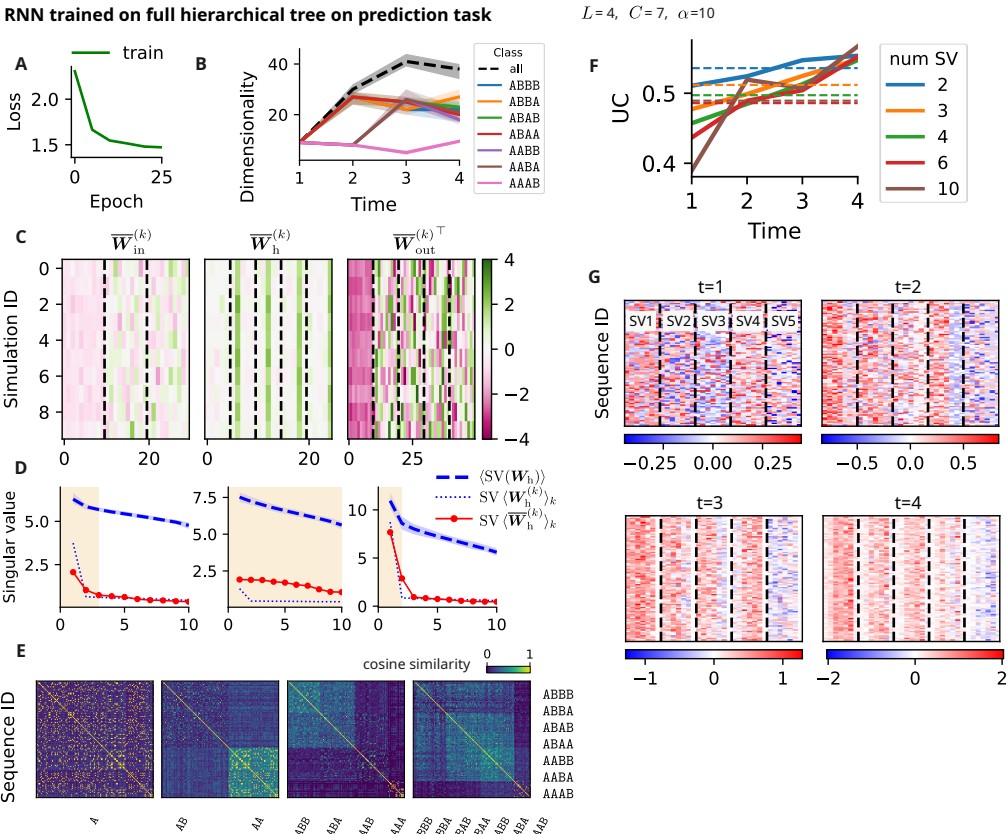

Figure S5: The results with "lazy" initialization regime hold in the "rich" case as well.

## H. PARAMETERS

We report here all the parameters required to reproduce the results. Table S1 recapitulates the parameters shared across all experiments in the manuscript, while Tables S2, S3 and S4 are specific to figures.

Table S1: Table of parameters shared across all experiments.

| Description | Size |
|---|---|
| **Network** | |
| Learning rate | 0.001 |
| Batch size | 1 |
| Initialization | $\mathcal{U}\left(-\frac{1}{\sqrt{N}}, \frac{1}{\sqrt{N}}\right)$ |
| Transfer function | ReLU |
| Loss function | Cross-Entropy |
| **Data** | |
| Number of unique letters per sequence $m$ | 2 |
| Size of alphabet $\alpha$ | 10 |

Table S2: Table of parameters for Fig. 1 of main text.

| Description | Size |
|---|---|
| **Network** | |
| Number of input neurons | 10 |
| Number of recurrent neurons $N$ (when not explicitly varied) | 128 |
| Number of output neurons | 4 |
| Number of training epochs | 41 |
| **Data** | |
| Sequence length $L$ (when not explicitly varied) | 6 |
| Sequence cue length $L_{cue}$ | 4 |
| Number of classes $C$ (when not explicitly varied) | 4 |
| Fraction of train sequences | 0.8 |
| Max number of class combinations $CC$ | 20 |
| Number of simulations with different initialization seeds | 5 |

Table S3: Table of parameters for Figs. 2 and 3 of main text.

| Description | Size |
|---|---|
| **Network** | |
| Number of input neurons | 10 |
| Number of recurrent neurons $N$ | 160 |
| Number of output neurons | 7 |
| Number of training epochs | 101 |
| **Data** | |
| Sequence length $L$ | 4 |
| Sequence cue length $L_{cue}$ | 4 |
| Number of classes $C$ | 7 |
| Fraction of train sequences | 1. |
| Number of class combinations $CC$ | 1 |
| Number of simulations with different initialization seeds | 100 |

Table S4: Table of parameters Fig. 4 of main text and Figs. S1, S2, and S3.

| Description | Size |
|---|---|
| Network | |
| Number of input neurons | 10 |
| Number of recurrent neurons $N$ (when not explicitly varied) | 160 |
| Number of output neurons (class, pred) | 5, 10 |
| Number of training epochs (class, pred) | 41, 101 |
| Data | |
| Sequence length $L$ (when not explicitly varied) | 9 |
| Sequence cue length $L_{cue}$ | 4 |
| Number of classes $C$ (when not explicitly varied) | 5 |
| Fraction of train sequences | 0.8 |
| Max number of class combinations $CC$ | 20 |
| Number of simulations with different initialization seeds | 5 |

