# OpenReview forum: "From sequences to schemas: How recurrent neural networks learn temporal abstractions"
_ICLR.cc/2026/Conference — Submitted to ICLR 2026_

### Official Review · Reviewer_3JvZ · 2025-10-15

**Soundness:** 3
**Presentation:** 3
**Contribution:** 3
**Rating:** 6
**Confidence:** 3

**Summary:**

The paper studies how RNNs learn and represent abstract temporal schemas from a sequence of time-varying inputs. By training RNNs on a sequence classification with binary tree structure, the authors found low-dimensional, linearly separable population dynamics and low-dimensional weights that support generalization to other sequences and tasks. The authors also show an interesting comparison with RNNs trained on the next-token prediction task, which didn’t display tree-structured neural activity as the sequence classification-trained RNNs do.

**Strengths:**

I’d like to start by thanking the authors for their thoughtful and well-executed work. This paper investigates how RNNs learn structured neural dynamics and connectivity patterns over sequences, a question that highlights the temporal regularities in the task and has been relatively underexplored in the literature. The authors conduct detailed and in-depth analyses of both RNN connectivity and dynamics, and also show how tree-like dynamical structures support generalization at the functional level. This provides a systematic view of how RNNs form temporal abstractions across multiple levels of analysis. I also found the experiment that extracts the dimensionality of low-rank structure in the recurrent weights by comparing the dimensionality of rotated averaged weights to that of the original averaged weights particularly clever. Overall, the analyses are well-motivated, carefully executed, and clearly interpreted. And I believe this work will be relevant for the computational and system neuroscience community.

**Weaknesses:**

1. To make prediction unambiguous the authors prune the tree heavily, which changes the data distribution and possibly the need for abstraction, and the observed lack of low-rank structure in prediction-based RNNs may be an artifact induced by the possibly unbalanced training data distribution. For example, can the authors compare the dimensionality / distribution of the training data for the prediction task, with that of the sequence prediction task?
2. The binary-tree, fixed-length, two-token-type setting is clean for analysis but narrow. It’s unclear how robust the low-rank abstraction is under (i) probabilistic transition of the tokens, (ii) variable-length sequences, (iii) larger alphabets and more than m=2 latent items. Can the authors comment on it?

**Questions:**

See weaknesses.

---

> ### Author Response · Authors · 2025-11-27
>
> Weakness 1:
>
> We would like to clarify that the issue arises from a misunderstanding about the data used in the comparison between classification and prediction tasks. The experiments in Sec.2 have been done using the full hierarchical tree. But in the experiments in Sec.3, where we compare the two tasks, the data for both tasks are identical, with sequences generated from the pruned tree in Fig.5A. We see that also in this pruned case, in the classification task, the low-rank recurrent connectivity (Fig.5E, red), and the reduction in the dimensionality of the dynamics (Fig.5D, red) holds, while the same cannot be said of the prediction task (Fig.5D and E, green).
>
> In relation to the question of the failure of the prediction net to learn a hierarchical schema, we have added in Fig.5G the similarity matrices between the hidden representations for the transfer experiments in Sec.3. We see there the dramatic difference induced by task objectives on the representations: in the prediction net trained from scratch, there is no apparent hierarchical structure in the similarity between hidden representations, in the transfer-and-retrain case, most of the hierarchical structure is destroyed by retraining. Finally, in the transfer-and-freeze case, identically to the classification task, the similarity exhibits a strong clustering, concomitant with the UC score reported in Fig.5F.
>
> We realize that this was poorly worded in the first version of the manuscript, so we edited Sec.3 of the text as well as Fig.5 to make this point clearer.
>
> Weakness 2:
>
> We thank the reviewer for these excellent suggestions that we are currently thinking about and/or implementing. But regarding the narrow scope, we argue that here our aim is not to survey different architectures, nor to make a general claim about the rich variety of temporal schemata present in sequences, but to understand in mechanistic detail and focus how a single recurrent net processes abstract temporal structures. We have reported results from two different task objectives (and to a lesser extent a third one, a reconstruction objective). Our aim is to reach a thorough understanding of how these different objectives shape internal representations, for which we have deemed it necessary to narrow down the generative process producing the sequences.
>
> (i) We fully agree that the extension of this work to more complex sequences, and especially probabilistic transitions or compositional sequences is a fascinating ongoing research effort we are invested in. Previous work by Yang et al [1] and Driscoll et al [2] have found dynamical motifs – recurring patterns of neural activity that implement specific computations through dynamics, such as attractors, decision boundaries and rotations, that are reused across different working memory tasks. We think that similar dynamical motifs or schema that are token-independent may implement recurring substructures that may then be flexibly reused, and our study is the first step towards this larger effort.
>
> (ii) We considered the fixed-length setting to focus on a minimal and interpretable class of abstract temporal structures relevant to foundational questions in neuroscience and cognitive science. In the case of algebraic patterns such as those used in our study (e.g. based on sequences of transitions between same/different tokens), length generalization is inherently ill-defined: extending beyond trained sequence lengths introduces ambiguity i.e. multiple longer continuations are possible, all equally valid (i.e. consistent with some latent relational structure). For example, a class defined by AAB could both be extended in multiple ways e.g. AABB or AABA. Thus, unlike formal grammars with strict production rules, there is no unique ‘correct’ extension, complicating the notion of out-of-distribution length generalization in this setting.
>
> (iii) We have run experiments with larger alphabets, but do not see any qualitative difference in the results presented.
> On the question of larger m, we are currently running simulations for m=3. While this is ongoing work at the time of the rebuttal, we may be able to provide some intuition. We think the equivalent of Fig.3D is more difficult to interpret: while for m=2 we intuitively understand the logic of the low-rank connectivity in terms of transitions to “same” or “different” token (which can be represented as + or - along one SV), for m>2 the way transitions are processed temporally quickly becomes much more complex. For instance, for m=3, transitions may still be interpreted as “same”, “different, and never seen before” and “different, and last seen X steps back”, but there may be many other ways to express these temporal relationships. We hope to provide concrete responses to these questions in future work.
>
> [1] Task representations in neural networks trained to perform many cognitive tasks.
>
> [2] Flexible multitask computation in recurrent networks utilizes shared dynamical motifs.

---

### Official Review · Reviewer_TPAB · 2025-10-29

**Soundness:** 2
**Presentation:** 3
**Contribution:** 2
**Rating:** 4
**Confidence:** 4

**Summary:**

This paper explores how RNNs learn temporal abstractions from sequences. This work uses synthetic sequence tasks generated from a binary branching hierarchy, and compared models that learn under different (classification vs. next-token prediction) objectives. The results show that RNNs can implicitly learn to encode the hierarchical structure of the sequences in the hidden representation. Further analysis shows that this hierarchical encoding is achieved through low-rank recurrent dynamics in the representations and learned weights.

**Strengths:**

- This work goes beyond representation analysis and offers a detailed understanding of how hierarchical representations are supported in the trained model. The rank-constrained baseline experiments are quite interesting.
- I appreciate that this work explores and compares different training objectives and how they may link to different model behavior.
- This work has the potential to offer interesting implications for analyzing temporal abstraction in biological networks.

**Weaknesses:**

- Some conclusions seem potentially over-claimed. For example, next-token prediction was linked to a failure in inducing similar representational dynamics, but the next-token task formulation significantly changed the data distribution, which could have greatly reduced the learning pressure to encode hierarchical class structures. The transfer effect from the model weights learned under the classification objective also seems relatively weak from Figure 4C.
- It seems that there might be some inconsistencies regarding what counts as "low-rank" or "effective compression". In some cases (e.g. Figure 2a), a dimensionality around 10 is defined to be reduced to low-rank representations over time steps, but other times (e.g. Figure 4d) a dimensionality of a little over 10 is described as weak compression. Results from rank-constrained training also seem to suggest that the truncated weight matrix can result in non-trivial generalization gap, which may suggest that additional dimensions are important for differentiating lower-level (close to leaf) classes.
- I think this work could be greatly enhanced if there is further understanding of the factors contributing to the effective learning of these hierarchical, low-rank representations -- e.g., whether they develop in tandem/before/after memorized predictions, whether they're seeded by architectural complexity (see below).

**Questions:**

- What was the motivation for initializing in the lazy learning regime as opposed to the rich learning regime? It might be interesting to compare if a rich learning regime more consistently/quickly leads to a hierarchical representation.
- What is the size of the hidden layer? Have you tried increasing/decreasing the hidden size and would you expect redundant representational resource (larger size) to potentially promote memorization and inhibit structure learning?
- I'm curious what exact generalization failures the rank-constrained models exhibit -- these errors seem like potentially really interesting to understand if the leading dimensions are preserving more high-level class differences as opposed to later dimensions, etc.

---

> ### Author Response · Authors · 2025-11-27
>
> Weakness1:
>
> We clarify that the issue arises from a misunderstanding about the data used in the comparison between classification and prediction tasks. The experiments in Sec.2 have been done using the full hierarchical tree. But in the experiments in Sec.3, where we compare the two tasks, the data for both tasks are identical, with sequences generated from the pruned tree in Fig.5A. We see that also in this pruned case, in the classification task, the low-rank recurrent connectivity (Fig.5E, red), and the reduction in the dimensionality of the dynamics (Fig.5D, red) holds, while the same cannot be said of the prediction task (Fig.5D and E, green).
>
> In relation to the failure of the prediction net to learn a hierarchical schema, we have added in Fig.5G the similarity matrices between the hidden representations for the transfer experiments in Sec.3. We see there the dramatic difference induced by task objectives on the representations: in the prediction net trained from scratch, there is no apparent hierarchical structure in the similarity between hidden representations, in the transfer-and-retrain case, most of the hierarchical structure is destroyed by retraining. Finally, in the transfer-and-freeze case, identically to the classification task, the similarity exhibits a strong clustering, concomitant with the UC score (Fig.5F), despite the heavily pruned generative tree.
>
> Weakness 2:
>
> A dimensionality of >10 is weak compression relative to what one would obtain from the classification net (compare green, yellow to red in Fig.5D). The dimensionality is largely set by the number of same/different transitions at every time-step and therefore depends on the number of classes the net has to learn: beyond t=5 (Fig.5D), the dimensionality stays constant all the way to the end of the sequence, because beyond this point, all sequences can be unambiguously classified. In Fig.2A, we have C=7 and we obtain a dimensionality of roughly d=7 towards the end of the sequence. In Fig.5D, C=5 and while the classification net gets to a dimensionality of d=5 beyond t=5, that of the prediction net is d=20.
>
> For more detail, please refer to the response to Question 3.
>
> Weakness 3:
>
> Please see responses to questions.
>
> Q1:
>
> We chose lazy initialization as a conservative test case: in this regime, low-rank structure is less likely to emerge unless driven by task constraints. That we observe it in lazy training strengthens the case that the low rank is a required outcome of the task. We have repeated the experiments in the rich regime and we find that all our results regarding low-rank dynamics and connectivity hold also in the rich regime (Supp Fig.S5).
>
> Q2:
>
> Most results have been obtained using a size of N=160. However, we have run experiments with nets of different sizes. In general, we find that larger nets form representations with higher UC score and with better generalization accuracy (Fig.3E). We have also tested the generality of our transfer results for nets of different sizes and we find that for large enough nets (N=128 and above), our results hold (Fig. S3, B and C).
>
> Q3:
>
> We do not think this interpretation is correct, and we have included Sec.2.3 further exploring the causal link between low-rank geometry and generalization. We report a series of experiments in which we
>
> 1) Train rank-constrained experiments and show that these can learn the task (ranks 3 and up).
>
> 2) Inactivate the first mode of the recurrent matrix and evaluate both performance and resulting representations. This manipulation dramatically affects the performance of the net (confusion matrices in Fig.4, B, bottom left) and highlights the role of the leading mode of the recurrent connectivity: to integrate past same/different transitions with current same/transitions at every timestep, separating classes along the binary branching tree as the sequence unfolds. With this dominant mode removed, the net cannot integrate same/different transitions at every timestep and can only rely on current transitions driven by inputs to differentiate classes. Therefore, at the end of the sequence, the errors it makes are all according to the very last transition: all classes ending in **AA are classified as ABAA and all classes ending in **AB are classified as ABAB. This can be seen more in detail via the projections of the hidden activity onto the dominant mode: in the unperturbed case, the integration of past transitions leads to different values of the projection for the different classes (Fig.4, D, left and Fig.S4) whereas this nuance is lost in the perturbed case (Fig.4, D, right and Fig.S4).
>
> As to the generalization gap between full-rank and rank-constrained experiments, we have also looked at the errors with rank-constrained experiments (see confusion matrices in Fig.4B, top left) and we indeed find that these do not occur randomly: they tend to occur between classes that have a higher similarity in their hidden layer representations (Fig.4C, left).

---

### Official Review · Reviewer_gRSD · 2025-10-30

**Soundness:** 2
**Presentation:** 4
**Contribution:** 2
**Rating:** 4
**Confidence:** 4

**Summary:**

This paper investigates how Recurrent Neural Networks (RNNs) learn abstract temporal structures, such as algebraic patterns (e.g., AAB, ABA), from sequential data. The authors compellingly demonstrate that a sequence classification task, where a label is provided only at the end of the sequence, successfully drives the emergence of low-dimensional, low-rank recurrent dynamics that mirror the hierarchical, tree-like structure of the generative process. This is supported by a thorough set of analyses, including PCA, weight matrix SVD, and ultrametric content. The interesting part is that this schema learning is task-dependent, as standard next-token prediction fails to form such structures. Finally, the learned low-rank weights provide a reusable scaffold that accelerates learning and improves generalization in prediction tasks.

**Strengths:**

The paper is well-written and clearly articulates its fundamental research question: how neural networks learn abstract temporal schemas.

- **Core Finding on Task-Dependence:** The paper's primary strength is the clear demonstration that abstract, low-rank representations emerge specifically from a global sequence classification task. This is compellingly contrasted with a local next-token prediction task, which fails to produce the same abstract structure.
- **Thorough Analysis:** This central claim is supported by a convincing and multi-faceted analysis of the classification network's internal dynamics. The use of PCA , SVD of the recurrent weights, and ultrametricity analysis collectively builds a strong case that the network learns a low-dimensional, hierarchical representation of the abstract sequence classes.

**Weaknesses:**

1. **Different data distributions in task comparison:** The paper's central claim is that the task objective (global classification vs. local prediction) dictates whether an abstract, low-rank representation emerges. However, the study may introduce a confound by training the two models on different data distributions. The classification network was trained on sequences from a full, branching tree, while the prediction network was trained on "unambiguous" sequences derived from an extreme pruning of that same tree. This makes it impossible to disentangle the effect of the task objective from the effect of the input data's structure. The prediction network may have failed to learn the hierarchical schema simply because its training data was explicitly pruned to remove that rich, branching structure.
2. **Inadequate evidence for transfer learning:** The authors conclude that the low-rank structure learned during classification serves as a "reusable scaffold" for the prediction task. However, the paper's own analysis demonstrates that this scaffold is only beneficial when its weights are retrained. Crucially, during this retraining, the dimensionality increases, the recurrent weights become high-rank, and the representation's ultrametric content drops. This suggests that while the scaffold provides a good initialization, the abstract structure itself is not "reused" to solve the new task, which undermines the paper's conclusion about its functional transferability.

**Questions:**

1. The transfer experiment where the classification weights are frozen results in performance at chance level (Fig 4C, red line). This is a surprisingly complete failure. Does this imply that the abstract representations in $h_t$ are not, by themselves, sufficient for the output weights of the prediction network to solve the task?
2. Given that the low-rank, high-ultrametricity structure is lost upon retraining, how did the authors distinguish between the "reusable scaffold" hypothesis and the alternative that the classification pre-training merely provided a "warm start" in the prediction loss landscape?
3. The transfer learning analysis could be strengthened by a more granular ablation study. For instance, the authors could test a condition where the transferred recurrent weights ($W_h$) are frozen to preserve the abstract scaffold, while the transferred input weights ($W_{in}$) are retrained.
4. The paper links generalization accuracy to the Ultrametric Content (UC) score (Fig 3E). However, the UC score never approaches a perfect 1.0, and the similarity matrices (Fig 3A) show clear off-diagonal blocks. The authors attribute this to "short-term memory effects". Could this also be interpreted as the network learning a structure that is not a perfect hierarchy, but a more complex graph or schema?

---

> ### Author Response · Authors · 2025-11-27
>
> Weakness 1:
>
> We clarify that the issue arises from a misunderstanding about the data used in the comparison between classification and prediction tasks. The experiments in Sec.2 have been done using the full hierarchical tree. But in the experiments in Sec.3, where we compare the two tasks, the data for both tasks are identical, with sequences generated from the pruned tree in Fig.5A. We see that also in this pruned case, in the classification task, the low-rank recurrent connectivity (Fig.5E, red), and the reduction in the dimensionality of the dynamics holds (Fig.5D, red), while the same cannot be said of the prediction task (Fig.5D and E, green).
>
> In relation to the question of the failure of the prediction net to learn a hierarchical schema, we have added in Fig.5G the similarity matrices between the hidden representations for the transfer experiments in Sec.3. We see there the dramatic difference induced by task objectives on the representations: in the prediction net trained from scratch, there is no apparent hierarchical structure in the similarity between hidden representations, in the transfer-and-retrain case, most of the hierarchical structure is destroyed by retraining. Finally, in the transfer-and-freeze case, identically to the classification task, the similarity exhibits a strong clustering, concomitant with the UC score reported in Fig.5F.
>
> We realize that this was poorly worded in the first version of the manuscript, so we edited Sec.3 of the text as well as Fig.5 to make this point clearer.
>
> Weakness 2:
>
> We agree that the nature of transfer deserves careful clarification. However, we think that the current findings are not contradictory but highlight a subtle and interesting point about how inductive biases shaped by prior training can facilitate learning, even if they do not persist structurally after fine-tuning.
> 1) We think the transfer-and-freeze experiment fails because the representational geometry learned for end-of-sequence classification is not directly compatible with token-by-token forecasting. A recurrent scaffold optimized for one task is not directly reusable for another without adaptation.
> 2) Transfer-and-retrain experiment succeeds, improving asymptotic generalization (Fig.5C) even though fine-tuning degrades the explicit low-rank structure (Fig.5E). This indicates that the class-trained scaffold serves as an optimization inductive bias, a structured starting point from which the prediction net can reach a better solution.
> 3) To test whether this benefit is driven by abstract structure or by generic pretraining, we did a new control experiment where nets were pretrained on a reconstruction task (autoencoder architecture) using the same sequence data (Supp Fig.S3, E and F). Transferring the recurrent weights of the encoder layer to the prediction recurrent net conferred an advantage in memorization; however, it did not result in a better asymptotic generalization performance, unlike classification pretraining (Supp Fig.S3, A). This supports our claim that although generic pretraining may facilitate faster learning, it is the abstract structure of the representations that lead to better generalization.
>
> Q1:
>
> Yes, when both input and recurrent weights are transferred and frozen, the output layer alone cannot adapt to learn the prediction task.
>
> Q2:
>
> See response to weakness above. Briefly:
> Autoencoder pretraining: faster memorization, no generalization benefit
> Classification pretraining: faster memorization and better asymptotic generalization
> Thus the inductive bias introduced by abstract structure, not a generic “warm-start”, drives transfer.
>
> Q3:
>
> We ran a finer-grained transfer experiment in which the recurrent weights are transferred and frozen while the input weights are transferred and retrained (brown curve in Supp Fig.S3). This produced results almost identical to fully frozen transfer (both input and recurrent weights). This suggests that the net cannot adapt its input weights to make use of the abstract, low-dimensional representations learned in the classification task and must reconfigure these.
>
> Q4:
>
> We think that the reviewer’s understanding is correct: representing the sequences as a binary branching tree (BBT) in a “forward” form –that is, higher order splits in the classes are based on the earlier transitions– as in Fig.3B is one way of representing the sequences. An alternative is a BBT in the “backward” form –obtained by splitting classes according to the last transitions first. The similarity matrix exhibits both the forward tree and a backward tree. The net settles into a compromise geometry between these two organizations, but the forward hierarchy is more salient, as seen by the stronger diagonal blocks in the similarity matrix (Fig.3A). We think this asymmetry arises because the net processes sequences in the forward direction: early inputs begin shaping the hidden state, while later transitions are folded into already-compressed representations.

---

### Official Review · Reviewer_tYhh · 2025-10-31

**Soundness:** 3
**Presentation:** 2
**Contribution:** 2
**Rating:** 4
**Confidence:** 4

**Summary:**

This paper investigates how recurrent neural networks (RNNs) develop internal representations of abstract schemas during training. The authors design an abstract sequence classification task in which a binary branching tree generates structured sequences mapped to different concrete sequential units. RNNs are trained to classify the schema label, which is provided only at the end of each sequence.
The authors found that that RNNs trained on this end-of-sequence categorization task develop linearly separable internal representations that retain a tree-like structure. The authors further analyze the learned dynamics across randomly initialized RNNs, showing that independently trained networks converge to similar low-dimensional representations, differing only by random rotations. This shared low-rank structure, they argue, accounts for the network’s behavior. Finally, the authors noticed that jointly initializing both input and recurrent weights from an RNN trained on the schema classification improved performance for a network instructed to do a prediction task.

**Strengths:**

* This paper raises interesting questions about how RNNs internalize abstract schemas and converge to shared low-dimensional geometries.
* The paper provides a clean experimental design probing how abstract schema in shaping internal representations.
* The paper report experiments and analysis in a very honest and systematic way, which I appreciate.
* The discovery of shared low-dimensional computational geometries across independently trained networks is intriguing and may shed light on representational alignment in recurrent systems.

**Weaknesses:**

The observations are intriguing and thought-provoking, yet the theoretical grounding and causal interpretation remain underdeveloped. It also remains unclear how the findings on this particular form of schema generalize to many other forms of abstract patterns in sequential data.

* Scattered Presentation. The paper reads as a collection of interesting but loosely connected observations rather than a unified theoretical account. One wonders about the causal link between low-rank geometry and generalization.

* Lack of causal analysis: Section 2.2, I found the tree-like measure, and the correlation insufficient to justify the causal claim as the title for this subsection. As it is unclear whether a interference on the low rank recurrent activities will distroy/enhance RNN's representation of sequence relational structure.

* This paper lacks discussion about a range of literature in analyzing low rank RNNs (See Maheswaranathan et al. 2019, Schuessler et al. 2020) in the neuroscience; schema learning in the human cognition (see Wu et al. 2025a); and the type of mechanistic analysis done on RNNs' internal representations suggesting the emergence of internal representations reflective of sequence structure (Wu et al. 2025b). See Ref.

Additiona comments:
* Figure 1 C: there is no shaded area


Moreover, the focus on RNNs constrains the broader relevance of the work, as the audience of this conference would be curious on other architecture types... i.e. the scope is quite narrow.

**Questions:**

1.  What level of processing in the RNN reflects more abstract structure? Do later recurrent layers encode higher-level schema abstractions, while earlier ones capture more local bindings (e.g., abb = CDD/CFF/ACC)?
2. How the low rank recurrent activities drvies relational structure of sequences, what would happen if artificially induce low rank activities in these RNNs?  The singular value decomposition (SVD) is used descriptively but its causal or interpretive value is unclear. Does it reveal meaningful components that contribute to generalization or interpretability?
3. I am a bit surprised that the next-token-prediction RNNs do not form hierarchical representations. I suspect it has something to do with the way sequences are introduced to the RNNs, and a meta-learning type of set up may encourage the hierarchical representation to emerge in character predicting RNNs.
4.  Figure 3E suggests a weak correlation between UC scores and generalization ability. However, from the figure it is not obvious to see the correlation. And the relationship is not statistically tested—please include hypothesis testing or correlation significance analysis to support the claim.
5. Transfer. The final section suggests that networks trained for next-token prediction benefit from weights learned in schema classification networks. What is the role of low rank components in transfer?


Reference:

Human sequence schema learning:

_Wu, S., Thalmann, M., & Schulz, E. (2025) Two types of motifs enhance human recall and generalization of long sequences. Communications Psychology_

Analyzing low rank RNN:

_Maheswaranathan, N., Williams, A. H., Golub, M. D., Ganguli, S., & Sussillo, D. (2019). Universality and individuality in neural dynamics across large populations of recurrent networks. Nature Neuroscience_

_Schuessler, F., Mastrogiuseppe, F., Dubreuil, A., Ostojic, S., & Barak, O. (2020). Dynamics of random recurrent networks with low-rank structure. Physical Review Research_

Analyzing internal representations of RNNs:

_Wu, S., Alaniz, S., Karthik, S., Dayan, P., Schulz, E., & Akata, Z. (2025). Concept-Guided Interpretability via Neural Chunking. Advances in Neural Information Processing Systems 2025_

---

> ### Author Response · Authors · 2025-11-27
>
> Weaknesses.
>
> We thank the reviewer for these fruitful comments, and we have now reorganized the Results to make the narrative more coherent. We have included a Sec.2.3 exploring the causal link between low-rank geometry and generalization, and included the suggested references.
>
> Q1:
>
> In our setup we do not have multiple layers, just a single recurrent layer. Here our aim is not to survey different architectures, but to understand in mechanistic detail how a single recurrent net processes abstract temporal structures. Within this layer, we find that abstraction emerges internally over time, early timesteps reflect local transitions, while later timesteps integrate over the history of accumulated same/different transitions, producing hierarchical relational structure.
>
> Q2:
>
> We have included Sec.2.3 further exploring the causal link between low-rank geometry and generalization. Briefly:
> 1) We train rank-constrained nets and show that they can learn the task and maintain hierarchical representations.
> 2) We then inactivate the first mode of the recurrent matrix and evaluate both performance and representations. This manipulation dramatically affects the performance of the net (confusion matrices in Fig.4, B, bottom left) and highlights the role of the leading mode of the recurrent connectivity: to integrate past same/different transitions with current same/transitions at every timestep, separating classes along the binary branching tree as the sequence unfolds. With this dominant mode removed, the net cannot integrate same/different transitions at every timestep and can only rely on current transitions driven by inputs to differentiate classes. Therefore, at the end of the sequence, the errors it makes are all according to the very last transition: all classes ending in **AA are classified as ABAA and all classes ending in **AB are classified as ABAB. This can be seen more in detail via the projections of the hidden activity onto the dominant mode: in the unperturbed case, the integration of past transitions leads to different values of the projection for the different classes (Fig.4, D, left and Fig.S4) whereas this nuance is lost in the perturbed case (Fig.4, D, right and Fig.S4).
>
> Q3:
>
> We would like to highlight two points:
> 1) Hierarchical representations are not necessarily expected, even in the classification task. Representing the sequences as a binary branching tree (BBT) in a “forward” form –that is, higher order splits in the classes are based on the earlier transitions– as in Fig.3B is one way of representing the sequences. An alternative is a BBT in the “backward” form –obtained by splitting classes according to the last transitions first. The similarity matrix exhibits both the forward tree and a backward tree. The net settles into a compromise geometry between these two ultrametric organizations, but the forward hierarchy is more salient, as seen by the stronger diagonal blocks in the similarity matrix (Fig.3A). This asymmetry arises because the net processes sequences in the forward direction: early inputs begin shaping the hidden state, while later transitions are folded into already-compressed representations. Thus, the earliest transitions exert the strongest influence on representational geometry, yielding a more forward-dominant hierarchy.
> 2) Prediction requires token-identity memory. Successful prediction requires the RNN to retain exact token identities, not just relational structure. This pushes the net toward high-dimensional solutions that preserve identity-specific information and hinders the compression necessary for relational abstraction. As a result, hierarchical structure does not emerge spontaneously, even though the architecture is identical.
>
> Q4:
>
> In Fig.3E, the correlation is indeed weak for small nets. But we have also reported a generalization accuracy that is larger for nets of size N=128 and above.
>
> Q5:
>
> To test whether transfer depends specifically on the abstract low-rank scaffold, we conducted additional experiments in which:
> 1) We trained rank-constrained nets on classification.
> 2) We transferred their recurrent weights to prediction nets.
> 3) We evaluated learning speed and generalization. We found that even ranks as low as 2 still yield a transfer advantage (Supp Fig.S3 D).
> 4) To test whether the benefits reflect abstract structure rather than pretraining, we did a control experiment: we trained an autoencoder to reconstruct sequences. We found that it learns token identity and positional structure but not the abstract same/different relational schema. When we transferred its encoder weights into the prediction net, we observed faster memorization but not improved generalization, in sharp contrast to class-pretrained nets, which improved both learning speed and asymptotic generalization. These new results now included (Supp Fig.S3 E & F) support the idea that the abstract low-rank mode, rather than pretraining, facilitates generalization in the prediction task.

---

### Author Response · Authors · 2025-12-02
**Summary of the revised manuscript**

Dear ACs,
We would like to summarize the key points of our rebuttal and the major changes to the manuscript.

1. Task comparison–data are matched, representational differences are task-driven
Three reviewers flagged as a primary weakness that classification and prediction networks were trained on different data distributions. This was a misunderstanding due to our wording, not actual experiments.
Sec.2 (classification only) uses the full tree. But Sec.3, where we compare classification and prediction, uses identical data for both tasks (pruned tree, Fig.5A). Even on this matched distribution, the classification yields low-rank recurrent connectivity (Fig.5E, red), and strong dimensionality compression (Fig.5D, red), while prediction does not (Fig.5D and E, green).
To make this unequivocally clear, we added Fig.5G which shows the similarity matrices of hidden representations for all Sec.3 conditions (the transfer experiments). These reveal that:

- Prediction from scratch: no hierarchical organisation
- Transfer + retrain: hierarchical structure inherited from classification is mostly destroyed.
- Transfer + freeze: hierarchical clustering reappears, matching classification, including high UC (Fig.5F).

These additions make clear that difference arise from task objective, not data distribution. Sec. 3 and Fig.5 have been revised accordingly.

2. Causal role of the low-rank recurrent structure
A reviewer asked for causal evidence. We have now added Sec.2.3, with new causal perturbation analyses. Briefly:

1) We train rank-constrained networks and show that they can learn the task (for ranks ≥ 3) and maintain hierarchical representations.
2) Lesioning the the first singular component of the recurrent matrix after training produces catastrophic and highly structured failure (Fig.4B): the network can no longer integrate past transitions; all errors collapse onto mainly the final transitions. This highlights the role of the leading mode of the recurrent connectivity in integrating the current and past same/different transitions at every timestep, separating classes along the binary branching tree as the sequence unfolds. With this dominant mode removed, the network can only rely on current transitions driven by inputs to differentiate classes. Therefore, at the end of the sequence, the errors it makes are all according to the very last transition. Projections of the hidden activity onto the dominant mode confirm the causal mechanism (loss of accumulated transition signal; Fig.4D and Fig.S4)

These results causally show that the leading singular component of the recurrent connectivity implements the token-independent integration of same/different transitions over time; the computational backbone for abstraction and generalization.

3.Transfer: abstract scaffold vs warm start
Another main concern was regarding the classification to prediction transfer results: whether transfer reflects reuse of abstract structure or merely a generic ‘warm start’.
To disambiguate, we pretrained networks on a reconstruction task (autoencoder architecture) using the same sequence data (Supp Fig.S3, E and F). Transferring the recurrent weights of the encoder layer to the prediction recurrent net provides faster memorization only, but no generalization benefit. In contrast, classification pretraining improves both learning speed and generalization in prediction. These new results are now added in Supp Fig.S3 E & F.

4. Why prediction does not yield hierarchical representations
We clarified two points:
- Even classification RNNs produce only an approximate hierarchy. The sequences can be presented as a binary branching tree (BBT) in a “forward” (higher order splits in the classes are based on the earlier transitions, Fig3B), as well as in the “backward” form (splitting classes according to the last transitions). The cosine similarity matrix exhibits both trees (Fig.3A). The network effectively settles into a compromise geometry between these two ultrametric organizations, with the forward structure more prominent due to causal processing of sequences.
- Prediction requires preserving token identity, not just relational structure, pushing the model toward higher dimensional, token-specific representations, inhibiting the compression that enables hierarchical abstraction.

5. Robustness and scope
We addressed several concerns about robustness:
- Lazy vs rich initialization: qualitative results unchanged (Supp. FigS5)
- Network size: our results are stable for sufficiently large networks (Supp Fig.S1, S2, S3, B and C). In particular, we find that in the large network regime, there is a positive correlation between how hierarchical the representations are, and how well the network generalizes (Fig.3E).
- Generative process underlying sequences: our goal is a mechanistic account for a specific, interpretable family of algebraic patterns; expansions to probabilistic or compositional structures are ongoing but beyond the present scope.

---

### Meta-Review · Area_Chair_SvD9 · 2026-01-05

**Summary:**

This paper investigates how RNNs learn temporal abstractions from sequences. The authors use synthetic sequence tasks generated from a binary branching hierarchy, and compared different objectives (next-token prediction and classification). The authors show that RNNs can learn to implicitly encode hierarichical structure, and that this is done via low-rank recurrent dynamics in the representations and learned weights.

Reviewers appreciated the overall motivation of the paper, and how the experiments were designed and analysed.

Reviewers, did, however, raise a number of concerns related to the presentation and analysis of the experiments:
- "Scattered presentation" where the experiments are interesting, but do not fit a cohesive story.
- Missing discussions in analysing low-rank RNNs
- Low-rank structure and transfer learning unclear.
- Further insights into how low-rank structure emerges.

Although the authors posted a thorough rebuttal, these concerns are difficult to address and assess holistically without another review of the paper, as it requires substantial changes to the paper, and therefore on the balance, the final decision is to reject the paper. Authors are encouraged to revise their submission and to submit it to another venue.

**Reviewer Concerns:**

The concerns listed above are still outstanding.

**Reviewer Scores:**

Reviewer 3JvZ would probably retain weak accept rating.
Remaining reviewers may have retained original weak reject rating (4), or some may have raised their score to weak accept (6).

---

### Decision · Program_Chairs · 2026-01-26

Reject